# Learning Representations via a Robust Behavioral Metric for Deep Reinforcement Learning

**Jianda Chen**
Nanyang Technological University, Singapore
jianda001@e.ntu.edu.sg

**Sinno Jialin Pan**
Nanyang Technological University, Singapore
sinnopan@ntu.edu.sg

## Abstract

Learning an informative representation with behavioral metrics is able to accelerate the deep reinforcement learning process. There are two key research issues on behavioral metric-based representation learning: 1) how to relax the computation of a specific behavioral metric, which is difficult or even intractable to compute, and 2) how to approximate the relaxed metric by learning an embedding space for states. In this paper, we analyze the potential relaxation and/or approximation gaps for existing behavioral metric-based representation learning methods. Based on the analysis, we propose a new behavioral distance, the RAP distance, and develop a practical representation learning algorithm on top of it with a theoretical analysis. We conduct extensive experiments on DeepMind Control Suite with distraction, Robosuite, and autonomous driving simulator CARLA to demonstrate new state-of-the-art results.

## 1 Introduction

Deep reinforcement learning (RL) aims to interactively learn an optimal policy from high-dimensional environmental observations or states in an end-to-end manner. In the literature, it has been demonstrated that a robust representation of states, which are task-relevant and invariant to task-irrelevant background information, is able to significantly speed up the RL process and makes the learned policy more generalizable. Therefore, representation learning has played a key role in Deep RL algorithms and attracts more and more attention in the RL community [4, 23, 17, 20].

Prior work on representation learning is focused on learning embeddings to represent states based on a reconstruction loss [26, 25, 13]. Though promising results have been reported on some RL application domains, policies learned with such representations usually fail to generalize well in a complex environment because minimizing a reconstruction loss may potentially introduce local (visual) features with task-irrelevant information. Another research direction is to apply data augmentation techniques to learn robust state representations for an RL agent [19, 28, 27]. While data augmentation-based approaches may be able to learn more robust feature representations, they do not take the characteristics of Markov Decision Processes (MDPs), e.g., reward signals and dynamics models, into consideration when learning representations. As a result, the learned representations may not be informative for learning an optimal policy for the task of interest. A third research direction is to construct some auxiliary tasks in addition to the prime RL task and learn state representations by learning all the tasks simultaneously [14, 2, 21]. These approaches can be considered as shaping the state representations implicitly by learning the auxiliary tasks, which do not have a guarantee for learning a better policy, especially when the auxiliary tasks fail to benefit the learning of the RL task.

Recently, behavioral metrics, such as the bisimulation metric and its variants [7, 8, 3], have been exploited in representation learning for deep RL. Behavioral metrics are originally proposed to measure the behavioral similarity between states in terms of rewards and dynamics models, e.g, state transition probabilities. The high-level idea is to learn state embeddings by preserving the behavioral

36th Conference on Neural Information Processing Systems (NeurIPS 2022).

similarity between states based on a specific behavioral metric [29, 1, 16, 4]. As behavioral metrics provide a theoretical bound on the difference between the outputs of value function of a pair of states, the learned representation enjoys a theoretical guarantee to capture behavioral structure in the environment for policy learning. However, as behavioral metrics are expensive or even intractable to compute, different approximation approaches and learning objectives have been proposed to make behavioral metric-based representation learning for RL agents more efficient [29, 1, 16, 4].

Though behavioral metric-based representation learning methods have achieved state-of-the-art results on some benchmark RL problems, they suffer from at least one of the following three issues: loss function mismatch, relaxation of dynamics model divergence and the $L_1/L_2$ norm limitation.

- **Loss function mismatch**. In behavioral metric-based representation learning, a loss function is to measure the difference between the distance between states on the embedding space and their behavioral metric. As behavioral metrics are expensive or even intractable to computers, approximation or relaxation are necessary. However, based on our analysis, state-of-the-art methods adopt improper approximations, which may introduce a bias and make the bound of the value function looser.

- **Relaxation of dynamics model divergence**. The bisimulation metric or its on-policy invariant is one of the most widely used behavioral metrics, which requires estimating the 1-Wasserstein distance between dynamics models. As the 1-Wasserstein distance is usually difficult or intractable to estimate, prior methods propose some relaxations to replace the estimation of the 1-Wasserstein, which may break some theoretical guarantees of the bisimulation metric.

- **The $L_1/L_2$ norm limitation**. The $L_1$ and the $L_2$ norms are commonly-used distances with zero self-distance. However, due to the two gaps mentioned above, the approximations or relaxations of behavior metrics in prior approaches are potentially non-zero self-distance. Using the $L_1$ or $L_2$ distance on embedding space is difficult to learn robust representation to preserve the behavioral similarity between states.

To address the aforementioned issues or gaps, in this paper, we first introduce a new behavior metric namely the *Reducing Approximation Gap* (RAP) distance, and then develop a practical approximation algorithm with consistency to its theoretical prototype. In this way, our algorithm is guaranteed to learn a robust state representation to capture the behavioral similarity between states. We conduct extensive experiments on DeepMind Control Suite (DMC) [24] with background distraction, Robosuite [30] and autonomous driving simulator CARLA [6] to demonstrate new state-of-the-art results compared with other behavior metric-based representation learning methods.

The contributions of this paper are two-fold: 1) we analyze the potential approximation gaps for existing behavioral metric-based representation learning approaches, and 2) we introduce a new behavior distance RAP and develop its practical approximation algorithm with theoretical guarantees.

## 2 Related Work

In RL, early research work has focused on learning state representations by designing and optimizing some auxiliary objectives in addition to the RL task of interest. For instance, Hafner et al. [11] propose to learn a dynamics model to predict future states with a reconstruction loss. Gelada et al. [9] aim to learn state representations by predicting a dynamics model and a reward function on an embedding space. Laskin et al. [17] apply contrastive learning with samples generated by the momentum encoder. Hansen et al. [12] predict an inverse model as an auxiliary task for representation learning. Another type of approaches aims to apply data augmentation techniques to improve representation learning. For instance, Laskin et al. [18] conduct an extensive study of data augmentation for deep RL with pixel-based input. Yarats et al. [28] adopt random crops on pixel-based input and add regularization terms on the Q-function objectives. Lee et al. [19] introduce convolutional neural networks with randomized parameters. Stooke et al. [23] use augmented samples for representation contrastive learning. In contrast to these works, our proposed method aims to encode state representations into a structural metric space based on behavioral metrics.

Recently, behavioral metric-based representation learning has attracted more and more attention in the RL community. For instance, Zhang et al. [29] aim to learn representations by approximating the bisimulation metric [7, 3] on an embedding space. Agarwal et al. [1] propose a behavioral

metric considering a distance between action distributions given states for representation learning. Kemertas and Aumentado-Armstrong [16] propose to improve the robustness of the representation learning method proposed in [29] by adding norm constraints on the embedding space and intrinsic rewards. Castro et al. [4] introduce a new behavioral distance and develop a sampling-based approach to preserve the behavioral similarity between states on the embedding space. Chen and Pan [5] propose to learn neural networks to approximate components in the bisimulation metric on the state embedding space.

## 3 Preliminaries

**Reinforcement Learning**  We consider a Markov Decision Process (MDP) defined by a tuple $\langle \mathcal{S}, \mathcal{A}, \mathcal{R}, P, \gamma \rangle$, where $\mathcal{S}$ is the high-dimensional state space, $\mathcal{A}$ is the action space, $P(\mathbf{s}'|\mathbf{s}, \mathbf{a})$ is the transition distribution that captures the probability of entering a next state $\mathbf{s}' \in \mathcal{S}$ given a current state $\mathbf{s} \in \mathcal{S}$ and an action $a \in \mathcal{A}$, $\mathcal{R} : \mathcal{S} \times \mathcal{A} \to \mathbb{R}$ is the reward function and $\gamma \in [0, 1)$ is the discount factor. In the sequel, we use $P_\mathbf{s}^a$ and $r_\mathbf{s}^a$ to denote $P(\cdot|\mathbf{s}, a)$ and $\mathcal{R}(\mathbf{s}, a)$, respectively. A policy $\pi(a|\mathbf{s})$ is a probability distribution over each action $a$ conditioned on a state $\mathbf{s}$. The value function $V^\pi : \mathcal{S} \to \mathbb{R}$ for a given policy $\pi$ at a state $\mathbf{s}$ is defined as the expected sum of discounted future rewards,

$$V^\pi(\mathbf{s}) = \mathop{\mathbb{E}}_{\substack{\mathbf{a}_t \sim \pi(\cdot|\mathbf{s}_t) \\ \mathbf{s}_{t+1} \sim P_{\mathbf{s}_t}^{\mathbf{a}_t}}} \left[ \sum_{t=0}^\infty \gamma^t r_{\mathbf{s}_t}^{\mathbf{a}_t} \mid \mathbf{s}_0 = \mathbf{s} \right].$$

The goal of reinforcement learning is to find an optimal policy $\pi^* = \arg\max_\pi V^\pi$ that maximizes the expected future rewards. In the scope of representation learning for deep RL, a state encoder $\phi : \mathcal{S} \to \mathbb{R}^n$ maps a high-dimensional state to a low-dimensional vector, with which a policy $\pi(a|\phi(\mathbf{s}))$ is learned.

**Bisimulation Metrics**  The bisimulation metric [7, 8] defines a pseudometric $d : \mathcal{S} \times \mathcal{S} \to \mathbb{R}$ to measure the behavioral distance between states. Recently, a variant of the bisimulation metric, *on-policy* bisimulation metric (or $\pi$-bisimulation metric) is proposed [3], which focuses on behaviors relative to a particular policy $\pi$. The $\pi$-bisimulation metric consists of a reward difference term and a Wasserstein distance in dynamics models between states.

**Theorem 3.1** ($\pi$-bisimulation metric [3]). *Let $\mathbb{M}$ be the set of all pseudometrics on space $\mathcal{S}$. A pseudometric transformation function $\mathcal{F}_B^\pi : \mathbb{M} \to \mathbb{M}$ is defined as,*

$$\mathcal{F}_B^\pi(d)(\mathbf{s}_i, \mathbf{s}_j) = |\mathbb{E}_{\mathbf{a}_i \sim \pi} r_{\mathbf{s}_i}^{\mathbf{a}_i} - \mathbb{E}_{\mathbf{a}_j \sim \pi} r_{\mathbf{s}_j}^{\mathbf{a}_j}| + \gamma W_1(d)(P_{\mathbf{s}_i}^\pi, P_{\mathbf{s}_j}^\pi) \tag{1}$$

*where $\mathbb{E}_{\mathbf{a}_i \sim \pi} r_{\mathbf{s}_i}^{\mathbf{a}_i} = \mathbb{E}_{\mathbf{a}_i \sim \pi(\cdot|\mathbf{s}_i)} r_{\mathbf{s}_i}^{\mathbf{a}_i}$, $P_{\mathbf{s}_i}^\pi = \mathbb{E}_{a \sim \pi(\cdot|\mathbf{s}_i)} P_{\mathbf{s}_i}^a$ and $W_1$ is the 1-Wasserstein distance. $\mathcal{F}_B^\pi$ has a least fixed point $d_B^\pi$ and $d_B^\pi$ is a $\pi$-bisimulation metric.*

The following theorem shows that the difference in value function is bounded by $d_B^\pi$.

**Theorem 3.2** (Value difference bound [3]). *Given states $\mathbf{s}_i$ and $\mathbf{s}_j$, and policy $\pi$, we have*

$$|V^\pi(\mathbf{s}_i) - V^\pi(\mathbf{s}_j)| \leq d_B^\pi(\mathbf{s}_i, \mathbf{s}_j). \tag{2}$$

**MICo distance**  The MICo distance [4] is a variant of the $\pi$-bisimulation metric, which measures the distribution distance between dynamics models by computing the distance between sampled next states from the dynamics models in order to avoid the computation of the Wasserstein distance.

**Theorem 3.3** (MICo distance [4]). *Let $\mathbb{M}$ be the space of distance function $d : \mathcal{S} \times \mathcal{S} \to \mathbb{R}$, if the MICo metric function $\mathcal{F}_M^\pi : \mathbb{M} \to \mathbb{M}$ is defined as,*

$$\mathcal{F}_M^\pi(d)(\mathbf{s}_i, \mathbf{s}_j) = |\mathbb{E}_{\mathbf{a}_i \sim \pi} r_{\mathbf{s}_i}^{\mathbf{a}_i} - \mathbb{E}_{\mathbf{a}_j \sim \pi} r_{\mathbf{s}_j}^{\mathbf{a}_j}| + \gamma \mathbb{E}_{\substack{\mathbf{s}_i' \sim P_{\mathbf{s}_i}^\pi \\ \mathbf{s}_j' \sim P_{\mathbf{s}_j}^\pi}} d(\mathbf{s}_i', \mathbf{s}_j'), \tag{3}$$

*then $\mathcal{F}_M^\pi$ has a unique fixed point $d_M^\pi$.*

**Theorem 3.4** (Value difference bound [4]). *Given states $\mathbf{s}_i$ and $\mathbf{s}_j$, and policy $\pi$, we have*

$$|V^\pi(\mathbf{s}_i) - V^\pi(\mathbf{s}_j)| \leq d_M^\pi(\mathbf{s}_i, \mathbf{s}_j). \tag{4}$$

# 4 Approximation Gaps in Behavioral Metric-based Representation Learning

The high-level idea of behavioral metric-based representation learning is to learn an embedding space such that after mapping states onto the embedding space, the behavioral similarity can be preserved. We denote the state encoder by $\phi_\omega : \mathcal{S} \to \mathbb{R}^n$ with parameters $\omega$ and the distance between states on the embedding space $\mathbb{R}^n$ by $\hat{d}(\phi_\omega(\mathbf{s}_i), \phi_\omega(\mathbf{s}_j))$, e.g., $\hat{d}$ can be the $L_2$ norm distance. The problem of representation learning in terms of $\omega$ can be cast as a minimization problem of the following expected squared difference or loss between the distance on the embedding space, $\hat{d}(\phi_\omega(\mathbf{s}_i), \phi_\omega(\mathbf{s}_j))$, and the corresponding behavior metric, $d^\pi(\mathbf{s}_i, \mathbf{s}_j)$, between any pair of states $\mathbf{s}_i$ and $\mathbf{s}_j$:

$$\mathcal{L}(\phi_\omega) = \mathbb{E}\left[\left(\hat{d}(\phi_\omega(\mathbf{s}_i), \phi_\omega(\mathbf{s}_j)) - d^\pi(\mathbf{s}_i, \mathbf{s}_j)\right)^2\right]. \tag{5}$$

To develop a practical algorithm to minimize the above objective, prior approaches adopt different approximation or relaxation strategies to make the resultant optimization problem computationally tractable. As discussed above, there are three common issues underlying most prior approaches, which introduce gaps between the practical algorithms and their theoretical prototypes, i.e., (5), namely loss function mismatch, relaxation of dynamics model divergence, and the $L_1$ or the $L_2$ norm limitation. We discuss these 3 gaps in detail in the following sections.

## 4.1 Loss Function Mismatch

In general, to learn the encoder $\phi_\omega$ by minimizing (5) is computationally intractable or expensive because of the computation of the behavior metric $d^\pi$. We take MICo (MICo-based representation learning [4]) as an example. By specifying the term $d^\pi(\mathbf{s}_i, \mathbf{s}_j)$ in (5) as the MICo distance defined in (3), the loss of MICo becomes,

$$\mathcal{L}(\phi_\omega) = \mathbb{E}\left[\left(\hat{d}(\phi_\omega(\mathbf{s}_i), \phi_\omega(\mathbf{s}_j)) - \left|\mathbb{E}_{\mathbf{a}_i \sim \pi} r_{\mathbf{s}_i}^{\mathbf{a}_i} - \mathbb{E}_{\mathbf{a}_j \sim \pi} r_{\mathbf{s}_j}^{\mathbf{a}_j}\right| - \gamma \mathbb{E}_{\substack{\mathbf{s}_i' \sim P_{\mathbf{s}_i}^\pi \\ \mathbf{s}_j' \sim P_{\mathbf{s}_j}^\pi}} \hat{d}(\phi_{\bar{\omega}}(\mathbf{s}_i), \phi_{\bar{\omega}}(\mathbf{s}_j))\right)^2\right], \tag{6}$$

where $\bar{\omega}$ is a copy of parameters for the target network. However, the reward expectations $\mathbb{E}_{\mathbf{a}_i \sim \pi} r_{\mathbf{s}_i}^{\mathbf{a}_i}$ and $\mathbb{E}_{\mathbf{a}_j \sim \pi} r_{\mathbf{s}_j}^{\mathbf{a}_j}$ in the 2nd term in (6) are computationally intractable and also difficult to estimate even based on sampling. Thus, Castro et al. [4] propose to approximate the loss in (6) with the following alternative loss,

$$\mathcal{L}(\phi_\omega) = \mathbb{E}_{\substack{\mathbf{a}_i \sim \pi, \mathbf{a}_j \sim \pi \\ \mathbf{s}_i' \sim P_{\mathbf{s}_i}^{\mathbf{a}_i}, \mathbf{s}_j' \sim P_{\mathbf{s}_j}^{\mathbf{a}_j}}}\left[\left(\hat{d}(\phi_\omega(\mathbf{s}_i), \phi_\omega(\mathbf{s}_j)) - \left|r_{\mathbf{s}_i}^{\mathbf{a}_i} - r_{\mathbf{s}_j}^{\mathbf{a}_j}\right| - \gamma \hat{d}(\phi_{\bar{\omega}}(\mathbf{s}_i), \phi_{\bar{\omega}}(\mathbf{s}_j))\right)^2\right]. \tag{7}$$

Note that in (7), the expectation operator on rewards is moved out of the absolute value operator of the difference between rewards to avoid the estimation of expectation over rewards, $\mathbb{E}_{\mathbf{a}_i \sim \pi} r_{\mathbf{s}_i}^{\mathbf{a}_i}$ and $\mathbb{E}_{\mathbf{a}_j \sim \pi} r_{\mathbf{s}_j}^{\mathbf{a}_j}$, and enable sampling more efficient. However, such a revision introduces a gap between solutions of minimizing (7) and (6), because in (7) the reference behavioral metric is no longer the MICo distance but the "shift" MICo distance defined as follows.

**Definition 4.1** (Shift MICo distance). The shift MICo distance function $\tilde{\mathcal{F}}_M^\pi$ is defined as

$$\tilde{\mathcal{F}}_M^\pi(d)(\mathbf{s}_i, \mathbf{s}_j) = \mathbb{E}_{\substack{\mathbf{a}_i \sim \pi \\ \mathbf{a}_j \sim \pi}} |r_{\mathbf{s}_i}^{\mathbf{a}_i} - r_{\mathbf{s}_j}^{\mathbf{a}_j}| + \gamma \mathbb{E}_{\substack{\mathbf{s}_i' \sim P_{\mathbf{s}_i}^\pi \\ \mathbf{s}_j' \sim P_{\mathbf{s}_j}^\pi}} d(\mathbf{s}_i', \mathbf{s}_j').$$

**Lemma 4.2** (Fixed-point). *The shift MICo distance function $\tilde{\mathcal{F}}_M^\pi$ has a unique fixed-point $\tilde{d}_M^\pi$.*

*Proof.* (Sketch) This can be proved by following the proof of Theorem 3.3 by using Banach's fixed-point theorem. A detailed proof can be found in Appendix A.1. □

The above lemma shows that the approximated distance on the embedding space $\hat{d}(\phi_\omega(\mathbf{s}_i), \phi_\omega(\mathbf{s}_j))$ still converges to the Shift MICo distance, $\tilde{d}_M^\pi$, by minimizing (7). However, as $\mathbb{E}_{\substack{\mathbf{a}_i \sim \pi \\ \mathbf{a}_j \sim \pi}} |r_{\mathbf{s}_i}^{\mathbf{a}_i} - r_{\mathbf{s}_j}^{\mathbf{a}_j}| \geq \left|\mathbb{E}_{\mathbf{a}_i \sim \pi} r_{\mathbf{s}_i}^{\mathbf{a}_i} - \mathbb{E}_{\mathbf{a}_j \sim \pi} r_{\mathbf{s}_j}^{\mathbf{a}_j}\right|$, we have the following theorem, whose proof can be found in Appendix A.2.

**Theorem 4.3** (Looser value difference bound). *Given states $\mathbf{s}_i$ and $\mathbf{s}_j$, and a policy $\pi$, we have*

$$|V^\pi(\mathbf{s}_i) - V^\pi(\mathbf{s}_j)| \leq d_M^\pi(\mathbf{s}_i, \mathbf{s}_j) \leq \tilde{d}_M^\pi(\mathbf{s}_i, \mathbf{s}_j). \tag{8}$$

Based on the above theorem, as the Shift MICo distance has looser value difference bound, it may be less relevant to the value function. As a result, the learned representation may not be able to encode the behavioral similarity between states accurately. Apart from MICo, other behavioral metric-based methods, which consist of the on-policy reward difference term, such as DBC [29, 16] and AMBS [5], also suffer from a similar approximation gap.

## 4.2   Relaxation of Dynamics Models Divergence

The $\pi$-bisimulation metric needs to compute the 1-Wasserstein distance $W_1$ between dynamics models to measure the distribution distance. In $\pi$-bisimulation metric-based representation learning, one can learn the encoder $\phi_\omega$ by minimizing the following loss,

$$\mathcal{L}(\phi_\omega) = \mathbb{E}\left[\left(\hat{d}(\phi_\omega(\mathbf{s}_i), \phi_\omega(\mathbf{s}_j)) - \left|\mathbb{E}_{\mathbf{a}_i \sim \pi} r_{\mathbf{s}_i}^{\mathbf{a}_i} - \mathbb{E}_{\mathbf{a}_j \sim \pi} r_{\mathbf{s}_j}^{\mathbf{a}_j}\right| - \gamma W_1(\hat{d})(P_{\phi_\omega(\mathbf{s}_i)}^\pi, P_{\phi_\omega(\mathbf{s}_j)}^\pi)\right)^2\right],$$

However, the 1-Wasserstein distance is computationally expensive or intractable. In DBC [29] the 2-Wasserstein distance $W_2$ is proposed to replace $W_1$, as $W_2$ has a convenient closed-form of a Gaussian distribution with respect to the $L_2$ distance. Specifically, the loss function of DBC with batched sampled transitions is defined as,

$$\mathcal{L}(\phi_\omega) = \mathbb{E}\left[\left(\hat{d}(\phi_\omega(\mathbf{s}_i), \phi_\omega(\mathbf{s}_j)) - \left|r_{\mathbf{s}_i}^{\mathbf{a}_i} - r_{\mathbf{s}_j}^{\mathbf{a}_j}\right| - \gamma W_2(\|\cdot\|_2)(\widehat{P}_{\phi_\omega(\mathbf{s}_i)}^{\bar{\pi}}, \widehat{P}_{\phi_\omega(\mathbf{s}_j)}^{\bar{\pi}})\right)^2\right],$$

where $\|\cdot\|_2$ is the $L_2$ norm, $\widehat{P}$ is a dynamics model on the representation space, and $\bar{\pi}$ is the expected policy output. The use of $W_2$ almost breaks all theoretical guarantees for the bisimulation metric. The existence of unique fixed-point in the bisimulation metric requires the continuity and monotonicity of $W_1$ with respect to $d$ [7]. The properties of continuity and monotonicity do not hold with $W_2$. Therefore there is no more guarantee about the fixed-point existence in DBC except that both the dynamics model and the policy $\pi$ are deterministic, in which case $W_2(d)$ degenerates to $d$ and Banach's fixed-point exists [16]. However, this assumption may be too strong to hold in practice.

Instead of using the family of Wasserstein distances, in MICo as shown in (3) the sample-based distribution divergence, $\mathbb{E}_{\mathbf{s}_i' \sim P_{\mathbf{s}_i}^\pi, \mathbf{s}_j' \sim P_{\mathbf{s}_j}^\pi} d(\mathbf{s}_i', \mathbf{s}_j')$, is used to measure the difference between dynamics models. This sample-based distribution divergence can be considered as a Łukaszyk-Karmowski metric. While a Wasserstein distance has zero self-distance, a Łukaszyk-Karmowski metric does not satisfy the identity of indiscernibles. As a result, the approximated distance on the learned embedding space based on the MICo distance, which involves a Łukaszyk-Karmowski metric to measure distance between dynamics models, may also suffer from the violation issue of the identity of indiscernibles.

## 4.3   Limitation of Using the $L_1$ or the $L_2$ Norm on the Embedding Space

As mentioned in Section 4.1, to avoid the expensive or intractable computation of the expectation over rewards, prior approaches, such as MICo, and DBC, use an alternative term to measure the reward difference between states, $\mathbb{E}_{\substack{\mathbf{a}_i \sim \pi \\ \mathbf{a}_j \sim \pi}} |r_{\mathbf{s}_i}^{\mathbf{a}_i} - r_{\mathbf{s}_j}^{\mathbf{a}_j}|$. In the following, we discuss the case that state $\mathbf{s}_i$ and $\mathbf{s}_j$ are identical.

**Lemma 4.4.** *If $\mathbf{s}_i = \mathbf{s}_j$, then $\mathbb{E}_{\substack{\mathbf{a}_i \sim \pi \\ \mathbf{a}_j \sim \pi}} |r_{\mathbf{s}_i}^{\mathbf{a}_i} - r_{\mathbf{s}_j}^{\mathbf{a}_j}| \geq 0$. The equality holds only if the reward function $r$ is constant w.r.t. action $a$ or the policy $\pi$ is deterministic.*

Note that in most RL tasks, a stochastic policy is widely used for exploration and a reward function is rarely constant w.r.t. actions. Therefore, in practice, $\mathbb{E}_{\substack{\mathbf{a}_i \sim \pi \\ \mathbf{a}_j \sim \pi}} |r_{\mathbf{s}_i}^{\mathbf{a}_i} - r_{\mathbf{s}_j}^{\mathbf{a}_j}| > 0$ for most RL tasks.

Besides, as mentioned in Section 4.2, a Łukaszyk-Karmowski metric measuring distance between dynamics models does not satisfy the identity of indiscernibles. Therefore, the behavioral metric, which is a sum of the reward difference term and the dynamics model distance term (no matter a Wasserstein distance or a Łukaszyk-Karmowski metric), can be greater than zero on a pair of identical states.

Note that both the $L_1$ and the $L_2$ norms satisfy the identity of indiscernibles, i.e. when $\mathbf{x}_i = \mathbf{x}_j$, $\|\mathbf{x}_i - \mathbf{x}_j\|_1 = \|\mathbf{x}_i - \mathbf{x}_j\|_2 = 0$. If we leverage the $L_1$ or the $L_2$ norm as the form of distance $\hat{d}$ on the embedding space to approximate the behavioral metric, then the identical states pairs has zero distance on embedding space. However, the regression target, i.e., the behavioral metric, is greater than zero on identical states. As a result, when minimizing regression loss between the $L_1$ / $L_2$ norm and the behavioral metric, the representations for similar states, especially identical ones, will be pushed apart in the embedding space.

Recently, AMBS [5] proposes to use neural networks to measure distance on state embedding space rather than using the $L_1$ or the $L_2$ norm. However, in this case, the learned "behavior metric" does not have theoretical support such as the fixed-point convergence guarantee. Besides, MICo proposes a new form of distance on the embedding space, which is a sum of angular distance between embeddings and the $L_2$ norm of the embeddings. Note that the proposed distance has non-zero self-distance.

## 5  The Proposed RAP Distance

We firstly propose a new behavioral metric to measure the state similarity without computing the Wasserstein distance between dynamics models in Section 5.1. The proposed behavioral metric namely the RAP distance enjoys theoretical properties such as fixed-point existence and a value difference bound. We then present a practical algorithm to learn the state encoder by approximating the RAP distance on the state embedding space in Section 5.2. Our algorithm uses the learned estimation of reward functions and dynamics models to provide distance approximation which is consistent to the behavioral metric. It addresses all the aforementioned approximation gaps and preserves the theoretical guarantee about the value function difference bound. Particularly, the approximation gap of loss function mismatch is addressed in Section 5.2, the relaxation of dynamics model divergence is addressed in Section 5.1, and the limitation of the $L_1$ or the $L_2$ norm on the embedding space is addressed in Section 5.3.

### 5.1  Definition and Properties of the RAP Distance

In order to avoid the high computational cost of $W_1$ or the approximation gap introduced by relaxation to $W_2$ as described in Section 4.2, we consider a distance measure between dynamics models $P^\pi_{\mathbf{s}_i}$ and $P^\pi_{\mathbf{s}_j}$ with sampling. To be specific, our on-policy behavioral distance is defined as follows.

**Definition 5.1** (the RAP distance). Let $\mathbb{M}$ be the space of distance function $d : \mathcal{S} \times \mathcal{S} \to \mathbb{R}$, the RAP distance function $\mathcal{F}^\pi_G : \mathbb{M} \to \mathbb{M}$ is defined as,

$$\mathcal{F}^\pi_G(d)(\mathbf{s}_i, \mathbf{s}_j) = \left| \mathbb{E}_{\mathbf{a}_i \sim \pi} \, r^{\mathbf{a}_i}_{\mathbf{s}_i} - \mathbb{E}_{\mathbf{a}_j \sim \pi} \, r^{\mathbf{a}_j}_{\mathbf{s}_j} \right| + \gamma \mathbb{E}_{\substack{\mathbf{a}_i \sim \pi \\ \mathbf{a}_j \sim \pi}} d(\mathbb{E}[s'_i], \mathbb{E}[s'_j]), \tag{9}$$

where $\mathbb{E}[s'_i] = \mathbb{E}_{s'_i \sim P^{\mathbf{a}_i}_{\mathbf{s}_i}}[s'_i]$ is the expectation value of next state over the dynamics model $P^{\mathbf{a}_i}_{\mathbf{s}_i}$.

We design the behavioral distance by measuring the expected states over dynamics models recursively, which removes the requirement of sampling on $P^{\mathbf{a}_i}_{\mathbf{s}_i}$ and $P^{\mathbf{a}_j}_{\mathbf{s}_j}$ but only performs sampling on the policy $\pi$. In practice, the expected next states are generated by a learnable approximated dynamics model as described in Section 5.2.

**Theorem 5.2.** $\mathcal{F}^\pi_G$ is a contraction mapping w.r.t. the $L_\infty$ norm and has a unique fixed-point $d^\pi_G$.

*Proof.* Let $d, d' \in \mathbb{M}$. We have

$$|\mathcal{F}^\pi_G(d)(\mathbf{s}_i, \mathbf{s}_j) - \mathcal{F}^\pi_G(d')(\mathbf{s}_i, \mathbf{s}_j)| = \left| \gamma \sum_{\mathbf{a}_i, \mathbf{a}_j} \pi(\mathbf{a}_i|\mathbf{s}_i)\pi(\mathbf{a}_j|\mathbf{s}_j)(d - d')(\mathbb{E}[s'_i], \mathbb{E}[s'_j]) \right| \leq \gamma \|d - d'\|_\infty.$$

Therefore, $\mathcal{F}^\pi_G$ is a contraction mapping w.r.t. the $L_\infty$ norm and there exists a unique fixed-point for $\mathcal{F}^\pi_G$ due to Banach's fixed-point theorem. This completes the proof. $\qquad\square$

Theorem 5.2 provides a convergence guarantee for the RAP distance that by iterating $\mathcal{F}^\pi_G$, distance $d$ will converge to the fixed-point $d^\pi_G$.

**Theorem 5.3** (Value function difference bound). *Given states $\mathbf{s}_i$ and $\mathbf{s}_j$, and a policy $\pi$, we have*

$$|V^\pi(\mathbf{s}_i) - V^\pi(\mathbf{s}_j)| \le d_G^\pi(\mathbf{s}_i, \mathbf{s}_j). \tag{10}$$

The proof can be found in Appendix A.3. Theorem 5.3 demonstrates that the RAP distance between states upper-bounds the difference of their states values.

## 5.2 Approximation of RAP

A straightforward way to learn a representation encoder to approximate the RAP distance on the embedding space is to minimize the loss in (5). However, the approximation gap of loss function mismatch as mentioned in Section 4.1 still exists if we relax the first term in (9) as $\mathbb{E}_{\substack{\mathbf{a}_i \sim \pi \\ \mathbf{a}_j \sim \pi}} |r_{\mathbf{s}_i}^{\mathbf{a}_i} - r_{\mathbf{s}_j}^{\mathbf{a}_j}|$. This leads to the learned metric having a looser value difference bound than the original behavioral metric as proved in Section 4.1. Here, we propose another alternative relaxation of $|\mathbb{E}_{\mathbf{a}_i \sim \pi} r_{\mathbf{s}_i}^{\mathbf{a}_i} - \mathbb{E}_{\mathbf{a}_j \sim \pi} r_{\mathbf{s}_j}^{\mathbf{a}_j}|$ to address this issue. Let $r_{\mathbf{s}}$ be a random variable over the action distribution defined by $p(r_{\mathbf{s}} = r_{\mathbf{s}}^{\mathbf{a}}) = \pi(\mathbf{a}|\mathbf{s})$. We first analyze the difference between $\mathbb{E}_{\substack{\mathbf{a}_i \sim \pi \\ \mathbf{a}_j \sim \pi}} \left[ |r_{\mathbf{s}_i}^{\mathbf{a}_i} - r_{\mathbf{s}_j}^{\mathbf{a}_j}|^2 \right]$ and $|\mathbb{E}_{\mathbf{a}_i \sim \pi} r_{\mathbf{s}_i}^{\mathbf{a}_i} - \mathbb{E}_{\mathbf{a}_j \sim \pi} r_{\mathbf{s}_j}^{\mathbf{a}_j}|^2$,

$$
\begin{aligned}
&\mathbb{E}_{\substack{\mathbf{a}_i \sim \pi \\ \mathbf{a}_j \sim \pi}} \left[ |r_{\mathbf{s}_i}^{\mathbf{a}_i} - r_{\mathbf{s}_j}^{\mathbf{a}_j}|^2 \right] - |\mathbb{E}_{\mathbf{a}_i \sim \pi} r_{\mathbf{s}_i}^{\mathbf{a}_i} - \mathbb{E}_{\mathbf{a}_j \sim \pi} r_{\mathbf{s}_j}^{\mathbf{a}_j}|^2 \\
&= \mathbb{E}_{\mathbf{a}_i \sim \pi} \left[ (r_{\mathbf{s}_i}^{\mathbf{a}_i})^2 \right] + \mathbb{E}_{\mathbf{a}_j \sim \pi} \left[ (r_{\mathbf{s}_j}^{\mathbf{a}_j})^2 \right] - 2\mathbb{E}_{\substack{\mathbf{a}_i \sim \pi \\ \mathbf{a}_j \sim \pi}} \left[ r_{\mathbf{s}_i}^{\mathbf{a}_i} r_{\mathbf{s}_j}^{\mathbf{a}_j} \right] - \left[ \mathbb{E}_{\mathbf{a}_i \sim \pi} r_{\mathbf{s}_i}^{\mathbf{a}_i} \right]^2 - \left[ \mathbb{E}_{\mathbf{a}_j \sim \pi} r_{\mathbf{s}_j}^{\mathbf{a}_j} \right]^2 + 2 \left[ \mathbb{E}_{\mathbf{a}_i \sim \pi} r_{\mathbf{s}_i}^{\mathbf{a}_i} \right] \left[ \mathbb{E}_{\mathbf{a}_j \sim \pi} r_{\mathbf{s}_j}^{\mathbf{a}_j} \right] \\
&= \mathbb{E}_{\mathbf{a}_i \sim \pi} \left[ (r_{\mathbf{s}_i}^{\mathbf{a}_i})^2 \right] - \left[ \mathbb{E}_{\mathbf{a}_i \sim \pi} r_{\mathbf{s}_i}^{\mathbf{a}_i} \right]^2 + \mathbb{E}_{\mathbf{a}_j \sim \pi} \left[ (r_{\mathbf{s}_j}^{\mathbf{a}_j})^2 \right] - \left[ \mathbb{E}_{\mathbf{a}_j \sim \pi} r_{\mathbf{s}_j}^{\mathbf{a}_j} \right]^2 - 2\mathbb{E}_{\substack{\mathbf{a}_i \sim \pi \\ \mathbf{a}_j \sim \pi}} \left[ r_{\mathbf{s}_i}^{\mathbf{a}_i} r_{\mathbf{s}_j}^{\mathbf{a}_j} \right] + 2 \left[ \mathbb{E}_{\mathbf{a}_i \sim \pi} r_{\mathbf{s}_i}^{\mathbf{a}_i} \right] \left[ \mathbb{E}_{\mathbf{a}_j \sim \pi} r_{\mathbf{s}_j}^{\mathbf{a}_j} \right] \\
&= Var[r_{\mathbf{s}_i}] + Var[r_{\mathbf{s}_j}] - 2Cov[r_{\mathbf{s}_i}, r_{\mathbf{s}_j}].
\end{aligned}
\tag{11}
$$

Since $r_{\mathbf{s}_i}$ and $r_{\mathbf{s}_j}$ are independent, $Cov[r_{\mathbf{s}_i}, r_{\mathbf{s}_j}] = 0$. Therefore, we have the reward difference term

$$|\mathbb{E}_{\mathbf{a}_i \sim \pi} r_{\mathbf{s}_i}^{\mathbf{a}_i} - \mathbb{E}_{\mathbf{a}_j \sim \pi} r_{\mathbf{s}_j}^{\mathbf{a}_j}| = \sqrt{ \mathbb{E}_{\substack{\mathbf{a}_i \sim \pi \\ \mathbf{a}_j \sim \pi}} \left[ |r_{\mathbf{s}_i}^{\mathbf{a}_i} - r_{\mathbf{s}_j}^{\mathbf{a}_j}|^2 \right] - Var[r_{\mathbf{s}_i}] - Var[r_{\mathbf{s}_j}] }$$

and the revised RAP distance at fixed-point as

$$
\begin{aligned}
d_G^\pi(\mathbf{s}_i, \mathbf{s}_j) = &\sqrt{ \mathbb{E}_{\substack{\mathbf{a}_i \sim \pi \\ \mathbf{a}_j \sim \pi}} \left[ |r_{\mathbf{s}_i}^{\mathbf{a}_i} - r_{\mathbf{s}_j}^{\mathbf{a}_j}|^2 \right] - Var[r_{\mathbf{s}_i}] - Var[r_{\mathbf{s}_j}] } \\
&+ \gamma \mathbb{E}_{\substack{\mathbf{a}_i \sim \pi \\ \mathbf{a}_j \sim \pi}} d_G^\pi (\mathbb{E}_{s_i' \sim P_{\mathbf{s}_i}^{\mathbf{a}_i}} [s_i'], \mathbb{E}_{s_j' \sim P_{\mathbf{s}_j}^{\mathbf{a}_j}} [s_j']).
\end{aligned}
\tag{12}
$$

In (12), we successfully move the expectation operator on rewards out of the absolute value operator without introducing any approximation gap caused by loss function mismatch. However, there are three issues that need to be further addressed: 1) the square root introduces new bias under sampling, 2) the variances $Var[r_{\mathbf{s}_i}]$ and $Var[r_{\mathbf{s}_j}]$ are intractable to compute, and 3) how to estimate the expected next states $\mathbb{E}_{s_i' \sim P_{\mathbf{s}_i}^{\mathbf{a}_i}} [s_i']$ and $\mathbb{E}_{s_j' \sim P_{\mathbf{s}_j}^{\mathbf{a}_j}} [s_j']$.

In order to reduce the bias issue introduced by the square root, we try to remove the square root in the loss of learning the RAP distance. We move the dynamics term in (12) to the left-hand side, then take square on both sides and get

$$\left( d_G^\pi(\mathbf{s}_i, \mathbf{s}_j) - \gamma \mathbb{E}_{\substack{\mathbf{a}_i \sim \pi \\ \mathbf{a}_j \sim \pi}} d_G^\pi (\mathbb{E}_{s_i' \sim P_{\mathbf{s}_i}^{\mathbf{a}_i}} [s_i'], \mathbb{E}_{s_j' \sim P_{\mathbf{s}_j}^{\mathbf{a}_j}} [s_j']) \right)^2 = \mathbb{E}_{\substack{\mathbf{a}_i \sim \pi \\ \mathbf{a}_j \sim \pi}} \left[ |r_{\mathbf{s}_i}^{\mathbf{a}_i} - r_{\mathbf{s}_j}^{\mathbf{a}_j}|^2 \right] - Var[r_{\mathbf{s}_i}] - Var[r_{\mathbf{s}_j}]. \tag{13}$$

Let $\hat{d}(\phi_\omega(\mathbf{s}_i), \phi_\omega(\mathbf{s}_j))$ be the approximated RAP distance between $\mathbf{s}_i$ and $\mathbf{s}_j$ parameterized by $\omega$. The loss for learning $\hat{d}(\phi_\omega(\mathbf{s}_i), \phi_\omega(\mathbf{s}_j))$ is to minimize the mean squared error between the left-hand side and the right-hand side in (13):

$$
\begin{aligned}
\mathcal{L} = \mathbb{E}\Bigg[ &\left( \hat{d}(\phi_\omega(\mathbf{s}_i), \phi_\omega(\mathbf{s}_j)) - \gamma \mathbb{E}_{\substack{\mathbf{a}_i \sim \pi \\ \mathbf{a}_j \sim \pi}} d_G^\pi (\mathbb{E}_{s_i' \sim P_{\mathbf{s}_i}^{\mathbf{a}_i}} [s_i'], \mathbb{E}_{s_j' \sim P_{\mathbf{s}_j}^{\mathbf{a}_j}} [s_j']) \right)^2 \\
&- \left( |r_{\mathbf{s}_i}^{\mathbf{a}_i} - r_{\mathbf{s}_j}^{\mathbf{a}_j}|^2 - Var[r_{\mathbf{s}_i}] - Var[r_{\mathbf{s}_j}] \right) \Bigg]^2.
\end{aligned}
\tag{14}
$$

Such a loss consists of a pair of squared difference terms as the regression target, which approximate the squared difference between distances of current states and distances of dynamics models, respectively.

The reward variance $Var[r_{\mathbf{s}}]$ is computationally intractable, but we can learn a neural network approximator to estimate it by assuming that the reward $r_{\mathbf{s}}$ on state $\mathbf{s}$ is Gaussian distributed. Let $r_\psi$ be the learned reward function approximation parameterized by $\psi$, which outputs a Gaussian distribution, $r_\psi(\mathbf{s}) = \{\widehat{\mu}(r_{\mathbf{s}}), \widehat{\sigma}(r_{\mathbf{s}})\}$, where $\widehat{\mu}(r_{\mathbf{s}})$ and $\widehat{\sigma}(r_{\mathbf{s}})$ are the mean and the standard deviation, respectively.

Similarly, in order to estimate the expected next states $\mathbb{E}_{s' \sim P_{\mathbf{s}}^{\mathbf{a}}}[s']$ with a neural network approximator on the embedding space, we learn a dynamics model $\widehat{P}$ taking input of state embedding $\phi(\mathbf{s})$ and action $\mathbf{a}$ and outputs a Gaussian distribution over the next state embedding, $\widehat{P}(\phi_\omega(\mathbf{s}), \mathbf{a}) = \{\widehat{\mu}(\widehat{P}_{\phi_\omega(\mathbf{s})}^{\mathbf{a}}), \widehat{\sigma}(\widehat{P}_{\phi_\omega(\mathbf{s})}^{\mathbf{a}})\}$, where $\widehat{\mu}(\widehat{P}_{\phi_\omega(\mathbf{s})}^{\mathbf{a}})$ and $\widehat{\sigma}(\widehat{P}_{\phi_\omega(\mathbf{s})}^{\mathbf{a}})$ are the mean vector and the standard deviation vector for predictive embeddings of the next state, respectively.

The RAP distance between expected next states, $d_G^\pi(\mathbb{E}[s_i'], \mathbb{E}[s_j'])$, can be approximated by $\hat{d}(\widehat{\mu}(\widehat{P}_{\phi_\omega(\mathbf{s}_i)}^{\mathbf{a}_i}), \widehat{\mu}(\widehat{P}_{\phi_\omega(\mathbf{s}_j)}^{\mathbf{a}_j}))$. The variance $Var[r_{\mathbf{s}}]$ can be replaced by the learned reward function output as $(\widehat{\sigma}(r_{\mathbf{s}}))^2$. By replacing the dynamics term and reward variance terms in (14), we propose the RAP loss defined as

$$
\mathcal{L}_{RAP}(\phi_\omega) = \mathbb{E}_\mathcal{D}\Bigg[\left(\hat{d}(\phi_\omega(\mathbf{s}_i), \phi_\omega(\mathbf{s}_j)) - \gamma \hat{d}(\widehat{\mu}(\widehat{P}_{\phi_{\bar\omega}(\mathbf{s}_i)}^{\mathbf{a}_i}), \widehat{\mu}(\widehat{P}_{\phi_{\bar\omega}(\mathbf{s}_j)}^{\mathbf{a}_j}))\right)^2
$$
$$
- \left(|r_{\mathbf{s}_i}^{\mathbf{a}_i} - r_{\mathbf{s}_j}^{\mathbf{a}_j}|^2 - (\widehat{\sigma}(r_{\mathbf{s}_i}))^2 - (\widehat{\sigma}(r_{\mathbf{s}_j}))^2\right)\Bigg]^2, \quad (15)
$$

where $\bar\omega$ is a copy of parameter with stop gradient and $\phi_{\bar\omega}$ is the encoder in the target network and $\mathcal{D}$ is the replay buffer or the set of data that RL algorithm, e.g. SAC [10], learns from. The loss (15) is trained over the transitions sampled from $\mathcal{D}$.

## 5.3 Explicit Form of Distance on the Embedding Space

As discussed in Section 4.3, the behavioral metric is usually with non-zero self-distance. Besides, the distance measured between the next-state distributions in the RAP distance is a Łukaszyk-Karmowski metric [22], which also has non-zero self-distance. Here, we adopt the approximation form of distance on the embedding space as proposed in MICo [4].

**Definition 5.4** (MICo approximation). Let $\hat{d}_G : \mathbb{R}^n \times \mathbb{R}^n \to \mathbb{R}$ be a distance function on representation space $\mathbb{R}^n$, $\mathbf{x}, \mathbf{y} \in \mathbb{R}^n$ and $k > 0$. $\hat{d}_G$ is defined as $\hat{d}_G(\mathbf{x}, \mathbf{y}) = \|\mathbf{x}\|_2^2 + \|\mathbf{y}\|_2^2 + k\theta(\mathbf{x}, \mathbf{y})$, where $\theta$ is angular function evaluating the absolute angle between vectors $\mathbf{x}$ and $\mathbf{y}$, and $k$ is a hyperparameter. In practice, we set $k = 10^{-5}$ for our method.

As the above form of distance produces non-zero self-distance, with $\hat{d}_G$ the approximation of the RAP distance will not push apart similar states on the embedding space.

**Lemma 5.5** (Non-zero self-distance). *The self-distance of $\hat{d}_G$ is not restrict to zero: $\hat{d}_G(\mathbf{x}, \mathbf{x}) = 2\|\mathbf{x}\|_2^2 \geq 0$. The equality holds only if all the elements of $\mathbf{x}$ are zero.*

## 5.4 Implementation

The network architecture of our method is shown in Figure 1. Our method is built upon SAC [10]. Actor and critic networks take input of state representation $\phi_\omega(\mathbf{s})$. The SAC objectives and the RAP regression loss $\mathcal{L}_{RAP}(\phi_\omega)$ are optimized jointly. More implementation details can be found in Appendix B. The source code is available at `https://github.com/jianda-chen/RAP_distance`.

Figure 1: Network architecture of our method.

# 6 Experiment

In this section, we evaluate the efficiency, robustness and generalization ability of the representation learned by our method on three RL benchmarks: 1) Distracting DeepMind Control Suite (DMC) [24], 2) Robosuite [30], and 3) CARLA [6]. These are all control tasks in continuous action spaces with visual input. We compare our method with several baselines and state-of-the-art methods including metric-based representation learning and data augmentation: 1) **SAC** [10], a baseline RL method for continuous control, 2) **MICo** [4], a sample-based behavioral metric-based representation learning method for RL, 3) **DBC** [29], a representation learning method by approximating the bisimulation metric with the $L_1$ norm, 4) **RobustDBC** [16], a DBC-styled method with intrinsic rewards and inverse dynamics, and 5) **DrQ** [28], an image augmentation method on pixel inputs RL.

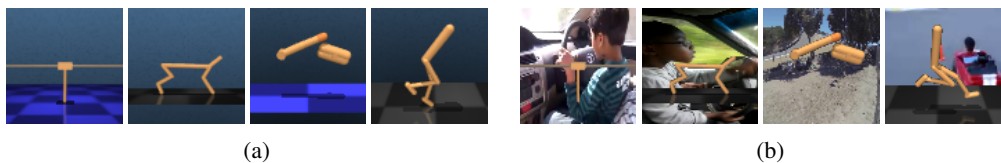

(a)                                                                                           (b)

Figure 2: Illustrations of observations in DMC: **(a)** original background setting; and **(b)** natural video background setting. From **left** to **right**: cartpole-swingup, cheetah-run, finger-spin, and walker-walk.

**Distracting DeepMind Control Suite**    To evaluate the generalization ability and robustness of our method, we perform experiments on 2 settings, original background and natural video background settings, in DMC [24]. We render $84{\times}84$ pixels as observation at each time step and stack frames as state. In the **original background** setting, we use the default background provided by DMC (as shown in Figure 2a). For the **natural video background** distracting setting, we follow [29] to replace the background with natural video sampled from Kinetics dataset [15] (as shown in Figure 2b). We sample 1000 continuous frames from the video dataset as background for training RL agents and evaluate agents on another 1000 continuous frames. The video background is considered as distraction to RL algorithm. For each setting, we train and evaluate on 4 tasks: cartpole-swingup, cheetah-run, finger-spin, and walker-walk. Each task is trained using 1 million environment steps.

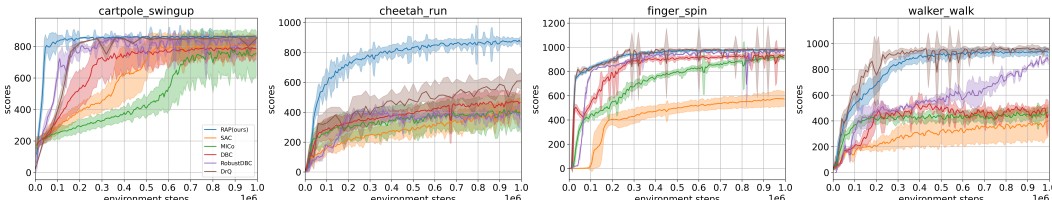

Figure 3: Experimental results on DMC with original background. Each curve is average on 3 runs.

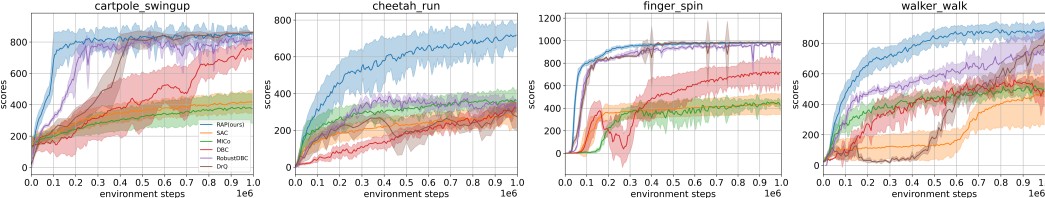

Figure 4: Results on natural video background settings in DMC. Each curve is average on 3 runs.

Figure 3 shows the training curves in original background settings. It confirms that our method RAP can accelerate the training and has comparable results to data augmentation method DrQ. Figure 4 shows the training curves in natural video background settings. Our method RAP outperforms DBC and MICo on all 4 tasks. It also converges to higher score than DrQ in 1 task and learns much faster in 3 tasks. The new SOTA results verify that our method of reducing the gaps in approximation of behavioral distance can improve the efficiency and robustness for deep RL and they also confirm that our method is able to learn generalizable state representation in complex environments.

**Robosuite** Robosuite [30] is a robotic simulator with various manipulation tasks based on MuJoCo engine. To evaluate the robustness of methods, we use the distraction settings for Robosuite by randomizing the colors of robot arms and table, the lighting source and luminance, and the camera position. The color, lighting and camera position are changed at the beginning of

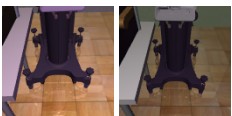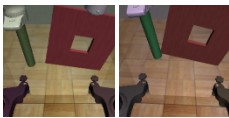

Figure 5: Illustrations in Robosuite. **Left**: Door Opening. **right**: Two Arm Peg-In-Hole.

an episode but are consistent within the episode. Our experiment is performed by controlling the Panda arms. The control rate is 20Hz and the episode length is 500 steps. We evaluate on two tasks: Door Opening and Two Arm Peg-In-Hole. Figure 5 shows illustrations of observations of two tasks. We compare our method with behavioral metric methods MICo and DBC, and data augmentation method DrQ. Table 1 shows the training scores in 500k environment steps. Our method RAP outperforms the others. The converge scores are around $7\times$ and $1.4\times$ compared to the second-best method in Door Opening and Two Arm Peg-In-Hole, respectively. It shows the robustness of our RAP in environments with random distractions.

Table 1: Experimental results on Robosuite trained with 500K environment steps.

| Task | RAP(Ours) | SAC | MICo | DBC | DrQ |
|---|---|---|---|---|---|
| Door Opening | **102.19 ± 26.11** | 8.84±9.89 | 6.06±6.57 | 3.15±3.54 | 14.63±19.57 |
| Two Arm Peg-In-Hole | **307.27±25.70** | 191.29±34.88 | 123.69±23.25 | 219.56±36.87 | 156.86±33.57 |

**CARLA** In order to validate the generalization ability and learning efficiency in natural scenarios, we construct experiments on an autonomous driving simulator CARLA [6], which provides 3D realistic on-world scenarios. The goal of this task is to control a vehicle driving as far as possible on a high-way map (Town 4) in 1000 time steps. The reward function followed [29] is designed to encourage driving far and avoid collisions with other vehicles. The observation is formed as $420 \times 84$ pixels of a 300 degrees ego-centric view, constructed by concatenating five cameras. In order to evaluate the generalization ability, we randomly sample a kind of weather in each episode starts. The weather (sunlight, rain, etc.), which affects the visual observation, can be consid-

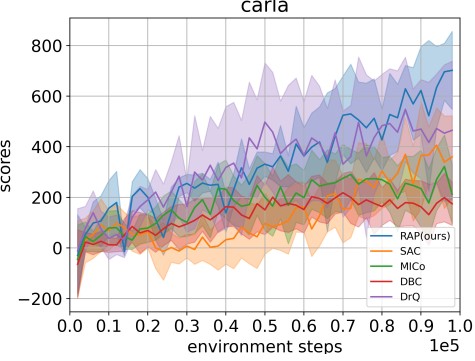

Figure 6: Training curves on CARLA.

ered as real-world distraction to RL agents. Figure 6 shows the comparison results of our RAP and other SOTA methods. Our method achieves comparable scores with data augmentation method DrQ, and its score is higher than behavioral metric DBC and MICo. It shows that our RAP is able to generalize well in tasks with real-world scenarios.

## 7 Conclusion

Representation learning is one of the most critical problems in high-dimensional deep RL. In this paper, we propose a new behavioral metric and a practical representation learning algorithm on top of the new behavioral metric for deep RL. We provide theoretical analysis for our proposed metric as well as the representation learning algorithm. We conduct empirical studies on multiple RL domains to verify the effectiveness of our proposed method.

## Acknowledgement and Funding Disclosure

This work is supported by the 2020 Microsoft Research Asia collaborative research grant.

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
