# A Proofs

## A.1 Proof of Lemma 4.2

**Lemma 4.2** (Fixed-point of shift MICo). *The "shift" MICo distance function $\tilde{\mathcal{F}}_M^\pi$ has a unique fixed-point $\tilde{d}_M^\pi$.*

*Proof.* The shift MICo distance function $\tilde{\mathcal{F}}_M^\pi$ is defined as,

$$\tilde{\mathcal{F}}_M^\pi(d)(\mathbf{s}_i, \mathbf{s}_j) = \mathbb{E}_{\substack{\mathbf{a}_i \sim \pi \\ \mathbf{a}_j \sim \pi}} |r_{\mathbf{s}_i}^{\mathbf{a}_i} - r_{\mathbf{s}_j}^{\mathbf{a}_j}| + \gamma \mathbb{E}_{\substack{\mathbf{s}_i' \sim P_{\mathbf{s}_i}^\pi \\ \mathbf{s}_j' \sim P_{\mathbf{s}_j}^\pi}} d(\mathbf{s}_i', \mathbf{s}_j').$$

Let $d, d' \in \mathbb{M}$. We have

$$\left| \tilde{\mathcal{F}}_M^\pi(d)(\mathbf{s}_i, \mathbf{s}_j) - \tilde{\mathcal{F}}_M^\pi(d')(\mathbf{s}_i, \mathbf{s}_j) \right|$$

$$= \left| \gamma \sum_{\mathbf{s}_i', \mathbf{s}_j'} \pi(\mathbf{a}_i|\mathbf{s}_i)\pi(\mathbf{a}_j|\mathbf{s}_j) P_{\mathbf{s}_i}^{\mathbf{a}_i}(\mathbf{s}_i') P_{\mathbf{s}_j}^{\mathbf{a}_j}(\mathbf{s}_j')(d - d')(s_i', s_j') \right|$$

$$\leq \gamma \|d - d'\|_\infty.$$

$\tilde{\mathcal{F}}_M^\pi$ is a contraction mapping w.r.t. the $L_\infty$ norm and there exists a unique fixed-point $\tilde{d}_M^\pi$ for $\tilde{\mathcal{F}}_M^\pi$ due to Banach's fixed-point theorem. This completes the proof. □

## A.2 Proof of Theorem 4.3

**Lemma A.1.** $\mathbb{E}_{\substack{\mathbf{a}_i \sim \pi \\ \mathbf{a}_j \sim \pi}} |r_{\mathbf{s}_i}^{\mathbf{a}_i} - r_{\mathbf{s}_j}^{\mathbf{a}_j}| \geq |\mathbb{E}_{\mathbf{a}_i \sim \pi} r_{\mathbf{s}_i}^{\mathbf{a}_i} - \mathbb{E}_{\mathbf{a}_j \sim \pi} r_{\mathbf{s}_j}^{\mathbf{a}_j}|$.

*Proof.* Since $r_{\mathbf{s}_i}^{\mathbf{a}_i}$ and $r_{\mathbf{s}_j}^{\mathbf{a}_j}$ are rewards which are scalars, we have $|r_{\mathbf{s}_i}^{\mathbf{a}_i} - r_{\mathbf{s}_j}^{\mathbf{a}_j}| \geq r_{\mathbf{s}_i}^{\mathbf{a}_i} - r_{\mathbf{s}_j}^{\mathbf{a}_j}$, and $|r_{\mathbf{s}_i}^{\mathbf{a}_i} - r_{\mathbf{s}_j}^{\mathbf{a}_j}| \geq r_{\mathbf{s}_j}^{\mathbf{a}_j} - r_{\mathbf{s}_i}^{\mathbf{a}_i}$ by symmetric. By taking the expectation over $\mathbf{a}_i$ and $\mathbf{a}_j$, we get,

$$\mathbb{E}_{\substack{\mathbf{a}_i \sim \pi \\ \mathbf{a}_j \sim \pi}} |r_{\mathbf{s}_i}^{\mathbf{a}_i} - r_{\mathbf{s}_j}^{\mathbf{a}_j}| \geq \mathbb{E}_{\substack{\mathbf{a}_i \sim \pi \\ \mathbf{a}_j \sim \pi}} \left[ r_{\mathbf{s}_i}^{\mathbf{a}_i} - r_{\mathbf{s}_j}^{\mathbf{a}_j} \right], \tag{16}$$

and

$$\mathbb{E}_{\substack{\mathbf{a}_i \sim \pi \\ \mathbf{a}_j \sim \pi}} |r_{\mathbf{s}_i}^{\mathbf{a}_i} - r_{\mathbf{s}_j}^{\mathbf{a}_j}| \geq \mathbb{E}_{\substack{\mathbf{a}_i \sim \pi \\ \mathbf{a}_j \sim \pi}} \left[ r_{\mathbf{s}_j}^{\mathbf{a}_j} - r_{\mathbf{s}_i}^{\mathbf{a}_i} \right]. \tag{17}$$

By combining (16) and (17), we have

$$\mathbb{E}_{\substack{\mathbf{a}_i \sim \pi \\ \mathbf{a}_j \sim \pi}} |r_{\mathbf{s}_i}^{\mathbf{a}_i} - r_{\mathbf{s}_j}^{\mathbf{a}_j}| \geq \left| \mathbb{E}_{\substack{\mathbf{a}_i \sim \pi \\ \mathbf{a}_j \sim \pi}} \left[ r_{\mathbf{s}_i}^{\mathbf{a}_i} - r_{\mathbf{s}_j}^{\mathbf{a}_j} \right] \right| = |\mathbb{E}_{\mathbf{a}_i \sim \pi} r_{\mathbf{s}_i}^{\mathbf{a}_i} - \mathbb{E}_{\mathbf{a}_j \sim \pi} r_{\mathbf{s}_j}^{\mathbf{a}_j}|.$$

□

**Lemma A.2** (Lifted MDP of MICo [4]). *The MICo metric function $\mathcal{F}_M^\pi$ is the Bellman operator for a Lifted MDP.*

*Proof.* Define the MDP for RL by a tuple $\langle \mathcal{S}, \mathcal{A}, \mathcal{R}, P, \gamma \rangle$. We consider a lifted MDP constructed by a tuple $\langle \hat{\mathcal{S}}, \hat{\mathcal{A}}, \hat{\mathcal{R}}, \hat{P}, \gamma \rangle$, where state space $\hat{\mathcal{S}} = \mathcal{S} \times \mathcal{S}$, action space $\hat{\mathcal{A}} = \mathcal{A} \times \mathcal{A}$, transition distribution $\hat{P}_{(\mathbf{s}_i, \mathbf{s}_j)}^{(\mathbf{a}_i, \mathbf{a}_j)} = P_{\mathbf{s}_i}^{\mathbf{a}_i} P_{\mathbf{s}_j}^{\mathbf{a}_j}$, and reward function $\hat{\mathcal{R}}((\mathbf{s}_i, \mathbf{s}_j)) = |\mathbb{E}_{\mathbf{a}_i \sim \pi} r_{\mathbf{s}_i}^{\mathbf{a}_i} - \mathbb{E}_{\mathbf{a}_j \sim \pi} r_{\mathbf{s}_j}^{\mathbf{a}_j}|$. The Bellman operator $T_M^{\hat{\pi}}$ under policy $\hat{\pi}(\mathbf{a}_i, \mathbf{a}_j|\mathbf{s}_i, \mathbf{s}_j) = \pi(\mathbf{a}_i|\mathbf{s}_i)\pi(\mathbf{a}_j|\mathbf{s}_j)$ is:

$$T_M^{\hat{\pi}}(d_M^\pi)((\mathbf{s}_i, \mathbf{s}_j)) = \sum_{(\mathbf{a}_i, \mathbf{a}_j)} \hat{\pi}(\mathbf{a}_i, \mathbf{a}_j|\mathbf{s}_i, \mathbf{s}_j) \sum_{(\mathbf{s}_i', \mathbf{s}_j')} \hat{P}_{(\mathbf{s}_i, \mathbf{s}_j)}^{(\mathbf{a}_i, \mathbf{a}_j)}(\mathbf{s}_i', \mathbf{s}_j') \left[ \hat{\mathcal{R}}((\mathbf{s}_i, \mathbf{s}_j)) + \gamma d_M^\pi(\mathbf{s}_i', \mathbf{s}_j') \right]$$

$$= |\mathbb{E}_{\mathbf{a}_i \sim \pi} r_{\mathbf{s}_i}^{\mathbf{a}_i} - \mathbb{E}_{\mathbf{a}_j \sim \pi} r_{\mathbf{s}_j}^{\mathbf{a}_j}| + \gamma \mathbb{E}_{\substack{\mathbf{s}_i' \sim P_{\mathbf{s}_i}^\pi \\ \mathbf{s}_j' \sim P_{\mathbf{s}_j}^\pi}} d_M^\pi(\mathbf{s}_i', \mathbf{s}_j')$$

$$= \mathcal{F}_M^\pi(d_M^\pi)(\mathbf{s}_i, \mathbf{s}_j).$$

□

**Lemma A.3** (Lifted MDP of shift MICo). *The shift MICo metric function $\tilde{\mathcal{F}}_M^\pi$ is the Bellman operator for a lifted MDP but with a different reward function.*

*Proof.* We construct the lifted MDP of shift MICo by a tuple $\langle \hat{\mathcal{S}}, \hat{\mathcal{A}}, \tilde{\hat{\mathcal{R}}}, \hat{P}, \gamma \rangle$ which is the same as the lifted MDP in Lemma A.2 except for the reward function $\tilde{\hat{\mathcal{R}}}((\mathbf{s}_i, \mathbf{s}_j), (\mathbf{a}_i, \mathbf{a}_j)) = |r_{\mathbf{s}_i}^{\mathbf{a}_i} - r_{\mathbf{s}_j}^{\mathbf{a}_j}|$. The Bellman operator $\tilde{T}_M^{\hat{\pi}}$ under policy $\hat{\pi}$ is:

$$
\tilde{T}_M^{\hat{\pi}}(\tilde{d}_M^\pi)((\mathbf{s}_i, \mathbf{s}_j))
$$
$$
= \sum_{(\mathbf{a}_i, \mathbf{a}_j)} \hat{\pi}(\mathbf{a}_i, \mathbf{a}_j | \mathbf{s}_i, \mathbf{s}_j) \sum_{(\mathbf{s}_i', \mathbf{s}_j')} \hat{P}_{(\mathbf{s}_i, \mathbf{s}_j)}^{(\mathbf{a}_i, \mathbf{a}_j)}(\mathbf{s}_i', \mathbf{s}_j') \left[ \tilde{\hat{\mathcal{R}}}((\mathbf{s}_i, \mathbf{s}_j), (\mathbf{a}_i, \mathbf{a}_j)) + \gamma \tilde{d}_M^\pi(\mathbf{s}_i', \mathbf{s}_j') \right]
$$
$$
= \mathbb{E}_{\substack{\mathbf{a}_i \sim \pi \\ \mathbf{a}_j \sim \pi}} |r_{\mathbf{s}_i}^{\mathbf{a}_i} - r_{\mathbf{s}_j}^{\mathbf{a}_j}| + \gamma \mathbb{E}_{\substack{\mathbf{s}_i' \sim P_{\mathbf{s}_i}^\pi \\ \mathbf{s}_j' \sim P_{\mathbf{s}_j}^\pi}} \tilde{d}_M^\pi(\mathbf{s}_i', \mathbf{s}_j')
$$
$$
= \tilde{\mathcal{F}}_M^\pi(\tilde{d}_M^\pi)(\mathbf{s}_i, \mathbf{s}_j).
$$

$\square$

**Lemma A.4.** *Consider an auxiliary MDP specified by a tuple $\langle \hat{\mathcal{S}}, \hat{\mathcal{A}}, \hat{\mathcal{R}}_\Delta, \hat{P}, \gamma \rangle$ where $\hat{\mathcal{R}}_\Delta((\mathbf{s}_i, \mathbf{s}_j), (\mathbf{a}_i, \mathbf{a}_j)) = |r_{\mathbf{s}_i}^{\mathbf{a}_i} - r_{\mathbf{s}_j}^{\mathbf{a}_j}| - |\mathbb{E}_{\mathbf{a}_i \sim \pi} r_{\mathbf{s}_i}^{\mathbf{a}_i} - \mathbb{E}_{\mathbf{a}_j \sim \pi} r_{\mathbf{s}_j}^{\mathbf{a}_j}|$. Let $V_\Delta^{\hat{\pi}}$ denote the value function of such MDP over policy $\hat{\pi}$, then*

$$
V_\Delta^{\hat{\pi}}((\mathbf{s}_i, \mathbf{s}_j)) \geq 0.
$$

*Proof.* The value function $V_\Delta^{\hat{\pi}}$ can be expanded as,

$$
V_\Delta^{\hat{\pi}}((\mathbf{s}_i, \mathbf{s}_j))
$$
$$
= \mathbb{E}_{\hat{\pi}} [\sum_t \gamma^t \hat{\mathcal{R}}_\Delta((\mathbf{s}_i^{(t)}, \mathbf{s}_j^{(t)}), (\mathbf{a}_i^{(t)}, \mathbf{a}_j^{(t)})) | \mathbf{s}_i^{(0)} = \mathbf{s}_i, \mathbf{s}_j^{(0)} = \mathbf{s}_j]
$$
$$
= \mathbb{E}_{\hat{\pi}} \left[ \sum_t \gamma^t \left( \mathbb{E}_{\substack{\mathbf{a}_i^{(t)} \sim \pi \\ \mathbf{a}_j^{(t)} \sim \pi}} \left| r_{\mathbf{s}_i^{(t)}}^{\mathbf{a}_i^{(t)}} - r_{\mathbf{s}_j^{(t)}}^{\mathbf{a}_j^{(t)}} \right| - \left| \mathbb{E}_{\mathbf{a}_i^{(t)} \sim \pi} r_{\mathbf{s}_i^{(t)}}^{\mathbf{a}_i^{(t)}} - \mathbb{E}_{\mathbf{a}_j^{(t)} \sim \pi} r_{\mathbf{s}_j^{(t)}}^{\mathbf{a}_j^{(t)}} \right| \right) \right| \mathbf{s}_i^{(0)} = \mathbf{s}_i, \mathbf{s}_j^{(0)} = \mathbf{s}_j \right].
$$

Due to Lemma A.1, $\mathbb{E}_{\substack{\mathbf{a}_i^{(t)} \sim \pi \\ \mathbf{a}_j^{(t)} \sim \pi}} \left| r_{\mathbf{s}_i^{(t)}}^{\mathbf{a}_i^{(t)}} - r_{\mathbf{s}_j^{(t)}}^{\mathbf{a}_j^{(t)}} \right| - \left| \mathbb{E}_{\mathbf{a}_i^{(t)} \sim \pi} r_{\mathbf{s}_i^{(t)}}^{\mathbf{a}_i^{(t)}} - \mathbb{E}_{\mathbf{a}_j^{(t)} \sim \pi} r_{\mathbf{s}_j^{(t)}}^{\mathbf{a}_j^{(t)}} \right| \geq 0$, therefore,

$$
V_\Delta^{\hat{\pi}}((\mathbf{s}_i, \mathbf{s}_j)) \geq 0.
$$

$\square$

**Theorem 4.3** (Looser value difference bound). *Given states $\mathbf{s}_i$ and $\mathbf{s}_j$, and a policy $\pi$, we have*

$$
|V^\pi(\mathbf{s}_i) - V^\pi(\mathbf{s}_j)| \leq d_M^\pi(\mathbf{s}_i, \mathbf{s}_j) \leq \tilde{d}_M^\pi(\mathbf{s}_i, \mathbf{s}_j).
$$

*Proof.* Due to Theorem 3.4, which has been proven by Castro et al. [4] in Proposition 4.8 of their paper, we already have the first inequality,

$$
|V^\pi(\mathbf{s}_i) - V^\pi(\mathbf{s}_j)| \leq d_M^\pi(\mathbf{s}_i, \mathbf{s}_j). \tag{18}
$$

Lemma A.2 shows that $d_M^\pi$ is the value function of lifted MDP $\langle \hat{\mathcal{S}}, \hat{\mathcal{A}}, \hat{\mathcal{R}}, \hat{P}, \gamma \rangle$. $d_M^\pi$ can be expanded as the sum of discounted future rewards,

$$
d_M^\pi(\mathbf{s}_i, \mathbf{s}_j) = \mathbb{E}_{\hat{\pi}} \left[ \sum_t \gamma^t \left( \left| \mathbb{E}_{\mathbf{a}_i^{(t)} \sim \pi} r_{\mathbf{s}_i^{(t)}}^{\mathbf{a}_i^{(t)}} - \mathbb{E}_{\mathbf{a}_j^{(t)} \sim \pi} r_{\mathbf{s}_j^{(t)}}^{\mathbf{a}_j^{(t)}} \right| \right) \right| \mathbf{s}_i^{(0)} = \mathbf{s}_i, \mathbf{s}_j^{(0)} = \mathbf{s}_j \right].
$$

Similarly, due to Lemma A.3, $\tilde{d}_M^\pi$ can be expanded as,

$$
\tilde{d}_M^\pi(\mathbf{s}_i, \mathbf{s}_j) = \mathbb{E}_{\hat{\pi}} \left[ \sum_t \gamma^t \left( \mathbb{E}_{\substack{\mathbf{a}_i^{(t)} \sim \pi \\ \mathbf{a}_j^{(t)} \sim \pi}} \left| r_{\mathbf{s}_i^{(t)}}^{\mathbf{a}_i^{(t)}} - r_{\mathbf{s}_j^{(t)}}^{\mathbf{a}_j^{(t)}} \right| \right) \right| \mathbf{s}_i^{(0)} = \mathbf{s}_i, \mathbf{s}_j^{(0)} = \mathbf{s}_j \right].
$$

Then we analyze the difference between $\tilde{d}_M^\pi$ and $d_M^\pi$. Since $\tilde{d}_M^\pi$ and $d_M^\pi$ are defined on the same policy $\hat\pi$ and dynamics model $\hat P$, we have

$$\tilde{d}_M^\pi(\mathbf{s}_i, \mathbf{s}_j) - d_M^\pi(\mathbf{s}_i, \mathbf{s}_j)$$

$$= \mathbb{E}_{\hat\pi}\left[\sum_t \gamma^t \left( \mathbb{E}_{\substack{\mathbf{a}_i^{(t)}\sim\pi \\ \mathbf{a}_j^{(t)}\sim\pi}} \left| r_{\mathbf{s}_i^{(t)}}^{\mathbf{a}_i^{(t)}} - r_{\mathbf{s}_j^{(t)}}^{\mathbf{a}_j^{(t)}} \right| - \left| \mathbb{E}_{\mathbf{a}_i^{(t)}\sim\pi} r_{\mathbf{s}_i^{(t)}}^{\mathbf{a}_i^{(t)}} - \mathbb{E}_{\mathbf{a}_j^{(t)}\sim\pi} r_{\mathbf{s}_j^{(t)}}^{\mathbf{a}_j^{(t)}} \right| \right) \middle| \mathbf{s}_i^{(0)} = \mathbf{s}_i, \mathbf{s}_j^{(0)} = \mathbf{s}_j \right]$$

$$= V_\Delta^{\hat\pi}((\mathbf{s}_i, \mathbf{s}_j)).$$

where $V_\Delta^{\hat\pi}((\mathbf{s}_i, \mathbf{s}_j))$ is the value function in Lemma A.4 and $V_\Delta^{\hat\pi}((\mathbf{s}_i, \mathbf{s}_j)) \geq 0$. Therefore,

$$d_M^\pi(\mathbf{s}_i, \mathbf{s}_j) \leq \tilde{d}_M^\pi(\mathbf{s}_i, \mathbf{s}_j). \tag{19}$$

By combining (18) and (19), we finally prove that,

$$|V^\pi(\mathbf{s}_i) - V^\pi(\mathbf{s}_j)| \leq d_M^\pi(\mathbf{s}_i, \mathbf{s}_j) \leq \tilde{d}_M^\pi(\mathbf{s}_i, \mathbf{s}_j).$$

$\square$

## A.3  Proof of Theorem 5.3

**Theorem 5.3** (Value function difference bound). *Given states $\mathbf{s}_i$ and $\mathbf{s}_j$, and a policy $\pi$, we have*

$$|V^\pi(\mathbf{s}_i) - V^\pi(\mathbf{s}_j)| \leq d_G^\pi(\mathbf{s}_i, \mathbf{s}_j). \tag{20}$$

*Proof.* We follow Castro et al. [4] (the proof of Proposition 4.8) to prove the value function difference bound. We will show that if $|V^\pi(\mathbf{s}_i) - V^\pi(\mathbf{s}_j)| \leq d(\mathbf{s}_i, \mathbf{s}_j), \forall \mathbf{s}_i, \mathbf{s}_j \in \mathcal{S}$, then $|V^\pi(\mathbf{s}_i) - V^\pi(\mathbf{s}_j)| \leq \mathcal{F}_G^\pi(d)(\mathbf{s}_i, \mathbf{s}_j)$. Suppose $|V^\pi(\mathbf{s}_i) - V^\pi(\mathbf{s}_j)| \leq d(\mathbf{s}_i, \mathbf{s}_j)$ holds, then

$$V^\pi(\mathbf{s}_i) - V^\pi(\mathbf{s}_j)$$

$$= \mathbb{E}_{\mathbf{a}_i\sim\pi} r_{\mathbf{s}_i}^{\mathbf{a}_i} - \mathbb{E}_{\mathbf{a}_j\sim\pi} r_{\mathbf{s}_j}^{\mathbf{a}_j} + \sum_{\mathbf{a}_i} \pi(\mathbf{a}_i|\mathbf{s}_i) \sum_{\mathbf{s}_i'} P_{\mathbf{s}_i}^{\mathbf{a}_i}(\mathbf{s}_i') V^\pi(\mathbf{s}_i') - \sum_{\mathbf{a}_j} \pi(\mathbf{a}_j|\mathbf{s}_j) \sum_{\mathbf{s}_j'} P_{\mathbf{s}_j}^{\mathbf{a}_j}(\mathbf{s}_j') V^\pi(\mathbf{s}_j')$$

$$\leq \left| \mathbb{E}_{\mathbf{a}_i\sim\pi} r_{\mathbf{s}_i}^{\mathbf{a}_i} - \mathbb{E}_{\mathbf{a}_j\sim\pi} r_{\mathbf{s}_j}^{\mathbf{a}_j} \right| + \mathbb{E}_{\substack{\mathbf{a}_i\sim\pi \\ \mathbf{a}_j\sim\pi}} \left( \sum_{\mathbf{s}_i'} P_{\mathbf{s}_i}^{\mathbf{a}_i}(\mathbf{s}_i') V^\pi(\mathbf{s}_i') - \sum_{\mathbf{s}_j'} P_{\mathbf{s}_j}^{\mathbf{a}_j}(\mathbf{s}_j') V^\pi(\mathbf{s}_j') \right).$$

We make an assumption that $\sum_{\mathbf{s}'} P_{\mathbf{s}}^{\mathbf{a}}(\mathbf{s}') V^\pi(\mathbf{s}') = V^\pi(\mathbb{E}_{\mathbf{s}'\sim P_{\mathbf{s}}^{\mathbf{a}}}[\mathbf{s}'])$. We make this assumption because we use the learned dynamics model to predict the next states, and the learned dynamics model is assumed as Gaussian distribution with small standard deviation. If this assumption holds, then we have

$$\left| \mathbb{E}_{\mathbf{a}_i\sim\pi} r_{\mathbf{s}_i}^{\mathbf{a}_i} - \mathbb{E}_{\mathbf{a}_j\sim\pi} r_{\mathbf{s}_j}^{\mathbf{a}_j} \right| + \mathbb{E}_{\substack{\mathbf{a}_i\sim\pi \\ \mathbf{a}_j\sim\pi}} \left( \sum_{\mathbf{s}_i'} P_{\mathbf{s}_i}^{\mathbf{a}_i}(\mathbf{s}_i') V^\pi(\mathbf{s}_i') - \sum_{\mathbf{s}_j'} P_{\mathbf{s}_j}^{\mathbf{a}_j}(\mathbf{s}_j') V^\pi(\mathbf{s}_j') \right)$$

$$= \left| \mathbb{E}_{\mathbf{a}_i\sim\pi} r_{\mathbf{s}_i}^{\mathbf{a}_i} - \mathbb{E}_{\mathbf{a}_j\sim\pi} r_{\mathbf{s}_j}^{\mathbf{a}_j} \right| + \mathbb{E}_{\substack{\mathbf{a}_i\sim\pi \\ \mathbf{a}_j\sim\pi}} \left( V^\pi(\mathbb{E}_{\mathbf{s}_i'\sim P_{\mathbf{s}_i}^{\mathbf{a}_i}}[\mathbf{s}_i']) - V^\pi(\mathbb{E}_{\mathbf{s}_j'\sim P_{\mathbf{s}_j}^{\mathbf{a}_j}}[\mathbf{s}_j']) \right)$$

$$\leq \left| \mathbb{E}_{\mathbf{a}_i\sim\pi} r_{\mathbf{s}_i}^{\mathbf{a}_i} - \mathbb{E}_{\mathbf{a}_j\sim\pi} r_{\mathbf{s}_j}^{\mathbf{a}_j} \right| + \mathbb{E}_{\substack{\mathbf{a}_i\sim\pi \\ \mathbf{a}_j\sim\pi}} d(\mathbb{E}_{\mathbf{s}_i'\sim P_{\mathbf{s}_i}^{\mathbf{a}_i}}[\mathbf{s}_i'], \mathbb{E}_{\mathbf{s}_j'\sim P_{\mathbf{s}_j}^{\mathbf{a}_j}}[\mathbf{s}_j'])$$

$$= \mathcal{F}_G^\pi(d)(\mathbf{s}_i, \mathbf{s}_j).$$

By symmetric,

$$V^\pi(\mathbf{s}_j) - V^\pi(\mathbf{s}_i) \leq \mathcal{F}_G^\pi(d)(\mathbf{s}_i, \mathbf{s}_j).$$

Therefore,

$$|V^\pi(\mathbf{s}_i) - V^\pi(\mathbf{s}_j)| \leq \mathcal{F}_G^\pi(d)(\mathbf{s}_i, \mathbf{s}_j).$$

We have an initial distance $d_0(\mathbf{s}_i, \mathbf{s}_j) = 2\max_{\mathbf{s}, \mathbf{a}} |r_{\mathbf{s}}^{\mathbf{a}}|/(1-\gamma)$ that satisfies $|V^\pi(\mathbf{s}_i) - V^\pi(\mathbf{s}_j)| \leq d(\mathbf{s}_i, \mathbf{s}_j)$, and $\mathcal{F}_G^\pi$ is contraction mapping on $d$. By repeatedly applying $\mathcal{F}_G^\pi$ on $d$, $d$ will eventually converge to the fixed-point $d_G^\pi$. Because for each iteration $\mathcal{F}_G^\pi(d)$ satisfies $|V^\pi(\mathbf{s}_j) - V^\pi(\mathbf{s}_i)| \leq \mathcal{F}_G^\pi(d)(\mathbf{s}_i, \mathbf{s}_j)$, the fixed-point $d_G^\pi$ satisfies

$$|V^\pi(\mathbf{s}_i) - V^\pi(\mathbf{s}_j)| \leq d_G^\pi(\mathbf{s}_i, \mathbf{s}_j). \tag{21}$$

$\square$

# B  Implementation Details

## B.1  Objectives

### B.1.1  Soft Actor-critic

Our approach is built upon Soft Actor-Critic (SAC) [10]. SAC is an actor-critic algorithm that optimizes a stochastic policy in an off-policy manner. We use the version of SAC that incorporates clipped double-Q trick. Let $Q_{\theta_i}$ denote the Q-networks where $i = 1, 2$, $Q_{\bar{\theta}_i}$ be the target Q-networks, and $\pi_\theta$ be the actor network. All Q-networks, target Q-networks and actor network take input of state representation $\phi_\omega(\mathbf{s})$ instead of state $\mathbf{s}$. The loss functions for the Q-networks are

$$\mathcal{L}_Q(Q_{\theta_i}, \phi_\omega) = \mathbb{E}_{(\mathbf{s},\mathbf{a},r,\mathbf{s}')\sim\mathcal{D}}\left[\left(Q_{\theta_i}(\phi_\omega(\mathbf{s}), \mathbf{a}) - r - \gamma V_{target}(\mathbf{s}')\right)^2\right],$$

where $\mathcal{D}$ is the replay buffer, and $V_{target}$ is the target value function. We stop gradients for $V_{target}$ and $V_{target}$ is defined as

$$V_{target}(\mathbf{s}') = \min_{j=1,2} Q_{\bar{\theta}_j}(\phi_{\bar{\omega}}(\mathbf{s}'), \mathbf{a}') - \alpha \log \pi_\theta(\mathbf{a}'|\phi_\omega(\mathbf{s}'))$$

where $\mathbf{a}' \sim \pi_\theta(\cdot|\phi_\omega(\mathbf{s}'))$ is the action at next state. The policy improvement is minimizing the following objective for actor network,

$$\mathcal{J}_\pi(\pi_\theta) = \mathbb{E}_{\mathbf{s}\sim\mathcal{D}}\left[\mathbb{E}_{\mathbf{a}\sim\pi_\theta(\cdot|\phi_\omega(\mathbf{s}))}\left[\alpha \log \pi_\theta(\mathbf{a}|\phi_\omega(\mathbf{s})) - \min_{i=1,2} Q_{\theta_i}(\phi_\omega(\mathbf{s}), \mathbf{a})\right]\right].$$

The loss function for $\alpha$ of SAC is,

$$\mathcal{J}_\alpha(\alpha) = \mathbb{E}_{\mathbf{s}\sim\mathcal{D}}\left[\mathbb{E}_{\mathbf{a}\sim\pi_\theta(\cdot|\phi_\omega(\mathbf{s}))}\left[\alpha \log \pi_\theta(\mathbf{a}|\phi_\omega(\mathbf{s})) - \alpha\bar{\mathcal{H}}\right]\right], \tag{22}$$

where $\bar{\mathcal{H}} \in \mathbb{R}$ is the target entropy and $\bar{\mathcal{H}} = -|\mathcal{A}|$ in practice.

### B.1.2  RAP Distance

We learn the reward function and dynamics model by neural network approximators, in order to approximate RAP distance. Let $r_\psi$ be the learned reward function parameterized by $\psi$, which outputs a Gaussian distribution, $r_\psi(\mathbf{s}) = \{\widehat{\mu}(r_\mathbf{s}), \widehat{\sigma}(r_\mathbf{s})\}$, where $\widehat{\mu}(r_\mathbf{s})$ and $\widehat{\sigma}(r_\mathbf{s})$ are the mean and the standard deviation, respectively. The loss function for learning $r_\psi$ is

$$\mathcal{L}_r(r_\psi) = \mathbb{E}_{(\mathbf{s},r,)\sim\mathcal{D}}\left[\left(\frac{r - \widehat{\mu}(r_\mathbf{s})}{2\widehat{\sigma}(r_\mathbf{s})}\right)^2\right].$$

We learn a dynamics model $\widehat{P}_\eta$ parameterized by $\eta$ to take input of state embedding $\phi(\mathbf{s})$ and action $\mathbf{a}$, and output a Gaussian distribution over the next state embedding, $\widehat{P}_\eta(\phi_\omega(\mathbf{s}), \mathbf{a}) = \{\widehat{\mu}(\widehat{P}^{\mathbf{a}}_{\phi_\omega(\mathbf{s})}), \widehat{\sigma}(\widehat{P}^{\mathbf{a}}_{\phi_\omega(\mathbf{s})})\}$, where $\widehat{\mu}(\widehat{P}^{\mathbf{a}}_{\phi_\omega(\mathbf{s})})$ and $\widehat{\sigma}(\widehat{P}^{\mathbf{a}}_{\phi_\omega(\mathbf{s})})$ are the mean vector and the standard deviation vector for the predictive next state embedding, respectively. The loss function for $\widehat{P}_\eta$ is

$$\mathcal{L}_P(\widehat{P}_\eta) = \mathbb{E}_{(\mathbf{s},\mathbf{a},\mathbf{s}')\sim\mathcal{D}}\left[\left(\frac{\phi_\omega(\mathbf{s}') - \widehat{\mu}(\widehat{P}^{\mathbf{a}}_{\phi_\omega(\mathbf{s})})}{2\widehat{\sigma}(\widehat{P}^{\mathbf{a}}_{\phi_\omega(\mathbf{s})})}\right)^2\right].$$

## B.2  Learning Algorithm

Algorithm shows the learning steps for learning SAC and RAP jointly.

---
**Algorithm 1** Leaning step for SAC and RAP
---
1: **Input:** Replay Buffer $\mathcal{D}$, Q network $Q_{\theta_j}$, actor $pi_\psi$, target Q network $Q_{\bar{\theta}_j}$, encoder $\phi_\omega$. reward function $r_\psi$, dynamics model $\widehat{P}$
2: Sample a batch with size $B$: $\{(\mathbf{s}_i, \mathbf{a}_i, r_i, \mathbf{s}'_i)\}_{b=1}^B \sim \mathcal{D}$
3: Update Q network by minimizing $\mathcal{L}_Q(Q_{\theta_i}, \phi_\omega)$
4: Update actor network by minimizing $\mathcal{J}_\pi(\pi_\theta)$
5: Update $\alpha$ by minimizing $\mathcal{J}_\alpha(\alpha)$
6: Update dynamics model $\widehat{P}_\eta$ by minimizing $\alpha_P \mathcal{L}_P(\widehat{P}_\eta)$
7: Update reward function $r_\psi$ by minimizing $\mathcal{L}_r(r_\psi)$
8: Update encoder $\phi_\omega$ by minimizing $\alpha_{RAP}\mathcal{L}_{RAP}(\phi_\omega)$
9: Softly update target Q network: $\bar{\theta}_j = \tau_Q \theta_j + (1 - \tau_Q)\bar{\theta}_j$
10: Softly update target encoder $\phi_{\bar{\omega}}$: $\bar{\omega} = \tau_\phi \omega + (1 - \tau_\phi)\bar{\omega}$
---

## B.3    Networks and Hyperparameters

We stack convolutional layers followed by a fully-connected layer as state encoder $\phi_\omega$. The encoder takes the input of a state, where 3 frames are stacked, and outputs a vector, the state representation $\phi_\omega(\mathbf{s})$. The detailed dimensions of each convolutional layer and fully-connected layer are described in Table 2. The Q-network consists of 3 stacked fully-connected layers with 1024 hidden dimensions, and takes the input of state representation $\phi_\omega(\mathbf{s})$ and action $\mathbf{a}$. The actor network takes the input of $\phi_\omega(\mathbf{s})$ and feeds it into three stacked fully-connected layers with 1024 hidden dimensions to output policy $\pi$. Both the approximated dynamics model $\widehat{P}$ and the approximated reward function $r_\psi$ are two-layers MLPs with 512 hidden dimensions. ReLU activation is used for every layer. We train the whole model on one NVIDIA A100 GPU. Other hyperparameters are listed in Table 2.

| Hyperparameter | DMC | Robosuite | CARLA |
|---|---|---|---|
| Episode length | 1000 | 500 | 1000 |
| Training steps | 1M | 500K | 100K |
| Replay buffer capacity | 1M | 100K | 100K |
| Batch size | 128 | 128 | 128 |
| Discount factor $\gamma$ | 0.99 | 0.99 | 0.99 |
| State dims | $9\times84\times84$ | $9\times128\times128$ | $9\times420\times84$ |
| Encoder conv kernels | [3,3,3,3] | [3,3,3,3] | [5, 3, 3, 3] |
| Encoder conv channels | [32,32,32,32] | [32,32,32,32] | [64,64,64,64] |
| Encoder conv strides | [2,1,1,1] | [2,1,1,1] | [2,2,2,2] |
| Encoder output dims | 100 | 100 | 100 |
| Optimizer | Adam | Adam | Adam |
| Q-networks learning rate | 5e-4 | 1e-3 | 1e-3 |
| Actor network learning rate | 5e-4 | 1e-3 | 1e-3 |
| Encoder learning rate | 5e-4 | 1e-3 | 1e-3 |
| Dynamics model learning rate | 5e-4 | 1e-3 | 1e-3 |
| Reward function learning rate | 5e-4 | 1e-3 | 1e-3 |
| log $\alpha$ learning rate | 1e-4 | 1e-4 | 1e-4 |
| $\tau_\phi$ | 0.05 | 0.05 | 0.05 |
| $\tau_Q$ | 0.01 | 0.01 | 0.01 |
| Target Q-network update frequency | 2 | 1 | 2 |
| Actor network update frequency | 2 | 1 | 2 |
| $\alpha_{RAP}$ | 0.5 | 0.5 | 0.5 |
| $\alpha_P$ | 1e-4 | 1e-4 | 1e-4 |
| Actor log std bound | [-10, 2] | [-10, 2] | [-10, 2] |

Table 2: Networks dimensions and hyperparameters.

## C  Additional Details of Experiments

### C.1  DeepMind Control Suite

Table 3 shows the action repeat for various DMC tasks.

| DMC Task | Action Repeat |
|---|---|
| Cartpole-Swingup | 8 |
| Cheetah-Run | 4 |
| Finger-Spin | 2 |
| Walker-Walk | 2 |

Table 3: Action repeat used for various tasks in DeepMind Control Suite.

### C.2  CARLA

Figure 7 shows an illustration of observation in CARLA. The observation is $420 \times 84$ pixels constructed by concatenating five cameras. The action repeat for CARLA is 8.

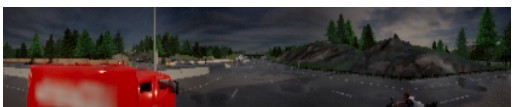

Figure 7: Illustration of observation in CARLA.

## D  Additional Experiments

### D.1  Ablation study on reward correction and learned dynamics

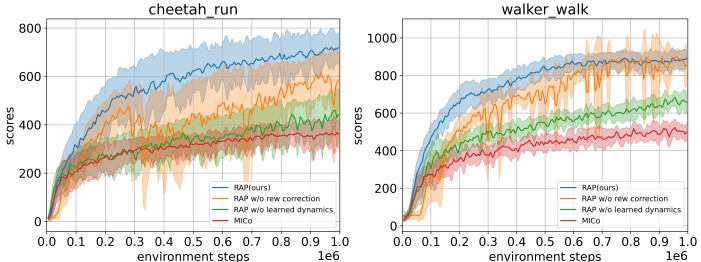

Figure 8: Ablation study results on DMC with natural video background.

We perform this ablation study in order to evaluate the effects of the reward correction and the learned Gaussian dynamics model. Figure 8 shows the experiment results of cheetah-run and walker-walk of DMC with natural video background setting. The curve **RAP w/o rew correction** is our method RAP with removing the reward variance correction. The curve **RAP w/o learned dynamics** is our method but measuring distances with next states that are sampled from MDP instead of a learned dynamics model mean/expectation. This curve can be considered as MICo with reward variance correction. Figure 8 shows that without reward variance correction our method becomes more unstable and gets worse results in cheetah-run. Without a learned dynamics model mean it will meet a larger performance drop but it is still better than MICo.

### D.2  Experiment on Simple 2D Highway

We add one more domain which is a 2D highway driving task to further evaluate our method and the baseline methods. The 2D highway task contains a straight road with 4 lanes and 50 other vehicles

running on. Figure 9 shows an illustration of 2D highway and Figure 10 shows the observation which is cropped to a square. The green vehicle is the vehicle that RL agent controls and blue vehicles are controlled by the environment. The RL agent learns to run as far as possible within 600 time steps.

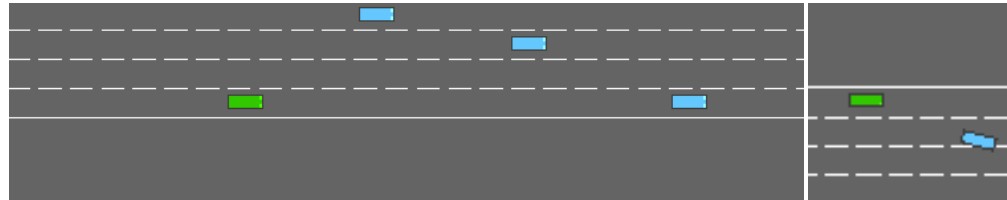

Figure 9: An illustration of 2D highway.

Figure 10: An example of observation in 2D highway task.

Figure 11 shows the learning curves at 500k steps. DBC, DrQ, and the proposed method RAP achieve similar scores. SAC and MICo are slightly worse than the other three. This 2D highway is a simple task so that they converge at around 100K to 200K steps.

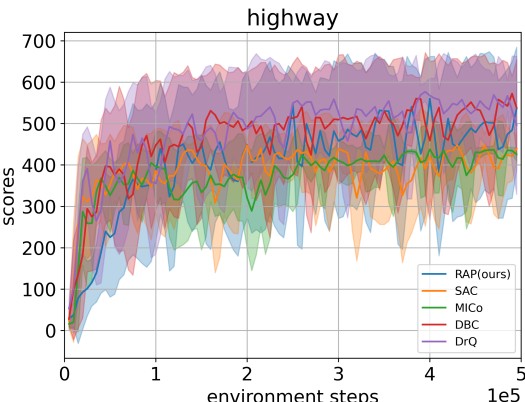

Figure 11: Experimental results on 2D Highway.

## E  Visualization of Representation Space

We show the t-SNE of the latent representation space of learned RAP, MICo, and DBC. We collect data from cheetah-run with different natural video backgrounds. Figure 12 shows the latent representation space where the data points are collected from four different video backgrounds and marked with four different colors. RAP and MICo can mix different video background which means they can prevent overfit to the background features, while DBC grouping the same color points into several clusters potentially captures the background task-irrelevant features. RAP maps the observations to a structural latent space but MICo maps to a circle latent space which may lose the structure.

Figure 13 shows the same representation spaces but each data point is colored according to the predicted Q-value. The darker blue indicates higher Q-value and lighter blue means lower Q-value. The representation space of RAP preserves the information for predicting Q-values as it gets dark blue in the bottom-left and light blue in the upper-right. MICo seems to have random Q-values with respect to the representation structure. DBC is likely to predict more extreme high/low Q-values.

Figure 14 shows some representations of the corresponding observations in the latent space. RAP can map similar observations to the nearby locations, while MICo and DBC map them far away from each other.

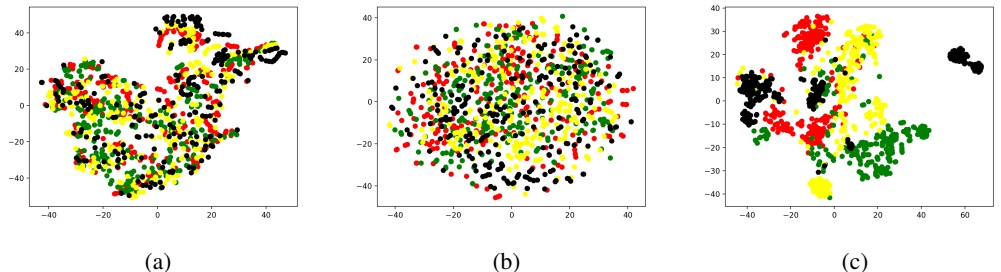

(a)          (b)          (c)

Figure 12: t-SNE of the latent representatiinon space of **(a)** RAP (ours), **(b)** MICo, and **(c)** DBC. Different colors indicate different videos in the background of DMC.

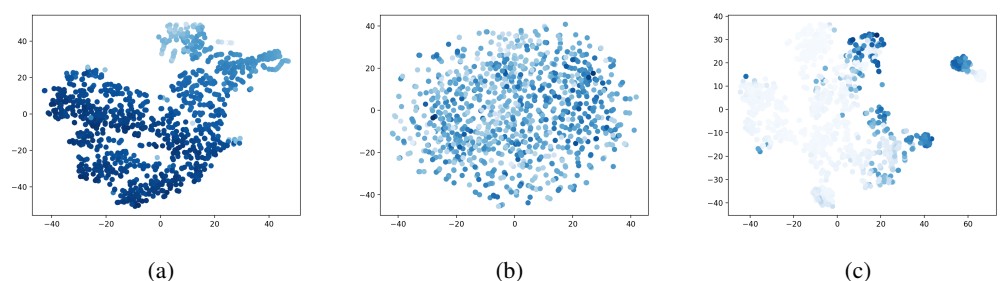

(a)          (b)          (c)

Figure 13: t-SNE of the latent representation space of **(a)** RAP (ours), **(b)** MICo, and **(c)** DBC. The color indicated the predicted Q-value (darker blue means a higher value).

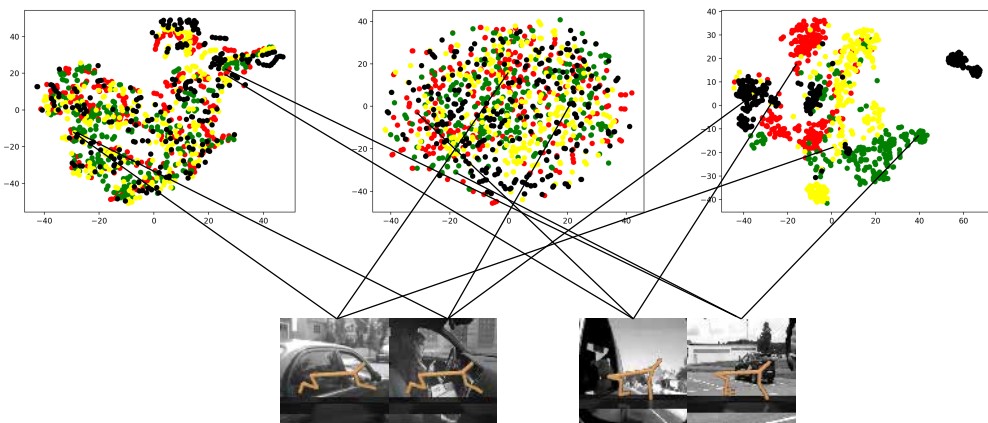

Figure 14: Comparison of the representations of similar observation pairs in latent space. From left to right: RAP, MICo, and DBC

## F  Explanation of Policy Dependence of RAP Loss

The loss (15) in Section 5.2 should be dependent on the policy. More precisely, the expectation is taken over the transitions sampled from a replay buffer $\mathcal{D}$. The loss (15) should be written as

$$
\begin{aligned}
\mathcal{L}_{RAP}(\phi_\omega) = \mathbb{E}_{\mathbf{s}_i,\mathbf{s}_j,r_{\mathbf{s}_i}^{\mathbf{a}_i},r_{\mathbf{s}_j}^{\mathbf{a}_j},\mathbf{a}_i,\mathbf{a}_j \sim \mathcal{D}} &\left( \left( \hat{d}(\phi_\omega(\mathbf{s}_i),\phi_\omega(\mathbf{s}_j)) - \gamma\hat{d}(\widehat{\mu}(\widehat{P}_{\phi_{\bar\omega}(\mathbf{s}_i)}^{\mathbf{a}_i}),\widehat{\mu}(\widehat{P}_{\phi_{\bar\omega}(\mathbf{s}_j)}^{\mathbf{a}_j})) \right)^2 \right. \\
&\left. - \left( |r_{\mathbf{s}_i}^{\mathbf{a}_i} - r_{\mathbf{s}_j}^{\mathbf{a}_j}|^2 - (\widehat{\sigma}(r_{\mathbf{s}_i}))^2 - (\widehat{\sigma}(r_{\mathbf{s}_j}))^2 \right)^2 \right).
\end{aligned}
\tag{23}
$$

The rewards $r_{\mathbf{s}_i}^{\mathbf{a}_i}$ and $r_{\mathbf{s}_j}^{\mathbf{a}_j}$ are sampled from the replay buffer, and the actions $\mathbf{a}_i$ and $\mathbf{a}_i$ that the rewards depend on are sampled from the policy when the rewards are generated from the MDP. Therefore this

|      | 1M   | 500K | 100K | 50K  | 10K  |
|------|------|------|------|------|------|
| 1M   | \    | 6.57 | 4.53 | 2.88 | 2.45 |
| 500K | 6.57 | \    | 3.43 | 5.76 | 5.21 |
| 100K | 4.53 | 3.43 | \    | 2.93 | 2.57 |
| 50K  | 2.88 | 5.76 | 2.93 | \    | 1.30 |
| 10K  | 2.45 | 5.21 | 2.57 | 1.30 | \    |

Table 4: Average differences of RAP distances among learned models with different replay buffer sizes.

loss function is dependent on the policy. The learning of the neural network approximated reward variance $\widehat{\sigma}$ is dependent on the data sampled from replay buffer, as described in Appendix B.1.2. This indicates that the learned reward function is also dependent on the policy.

If we use on-policy RL algorithms, then $\mathcal{D}$ would be the set of transitions sampled from the current policy. However, off-policy RL algorithms have better sample efficiency, especially in control tasks. In this paper, we build our method upon SAC which is also an off-policy algorithm. The underlying policy in the replay buffer may not be identical to the current policy. We want to verify that even though the learning of RAP is not dependent on the "truly" on-policy, our approximation is not far away from on-policy and is enough to learn good representations.

We try to reduce the replay buffer capacity. If the replay buffer is small enough, the RL algorithm would become on-policy. However, the smaller replay buffer usually leads to worse performance. We reduce the replay buffer capacity from 1 million to 500K, 100K, 50K, and 10K. Figure 15 shows the learning curves of different replay buffer capacities. It shows that smaller replay buffer sizes lead to lower performance and slower convergence.

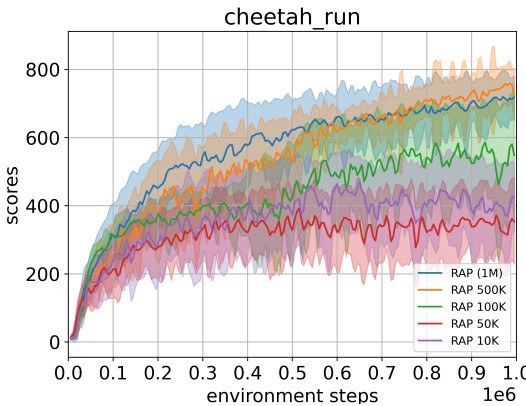

Figure 15: Experiment results in comparison of smaller replay buffer sizes.

We generate trajectories (in total 1K steps) by the optimal policy which is our RAP policy. Then we measure the RAP distance in between the 1K observations by using different models learned by different replay buffer sizes, i.e. 500K, 100K, 50K and 10K. We compare the distances among different replay buffer sizes and show the average differences in Table 4. The differences of distances stay on the same scale. The 1M version which is the proposed method is not far away from the models learned with other buffer sizes. Such difference/variance may be generated by the randomly initialized weights of the neural networks.