# OpenReview forum: "Learning Representations via a Robust Behavioral Metric for Deep Reinforcement Learning"
_NeurIPS.cc/2022/Conference — NeurIPS 2022 Accept_

### Official Review · Reviewer_nvyx · 2022-06-20

**Rating:** 6
**Confidence:** 4
**Soundness:** 2 fair
**Presentation:** 3 good
**Contribution:** 3 good

**Summary:**

This paper is about behavioral metrics and state representation. This paper proposes a behavioral distance, the RAP distance, and develops a practical representation learning algorithm on top of it. Experiments on DeepMind Control Suite with distraction, Robosuite, and autonomous driving simulator CARLA demonstrate promising results.

**Questions:**

In Equation 9, d is defined as a distance from a space of distance functions.
- Can you confirm that some distance functions d wouldn't make a lot of sense?
- Theorem 5.2 and theorem 5.3 uses the l_\infty loss but then in Section 5.2, a different distance function is used. How are the theorems useful for the practical algorithms?


**Limitations:**

As acknowledged by the authors, the limitations of the work are not discussed: "Did you describe the limitations of your work? [No]".

In addition to the lack of connexion between the theorems and the actual algorithm due to the metric d, an important characteristic/potential limitation of the RAP distance is that it depends on the policy. The explicit dependance on the policy is removed in Equation 13, why? How do you train the algorithm in practice given that samples might not be all obtained with one given known policy?

**Strengths And Weaknesses:**

Post rebuttal: I increase my score to 6 (see discussion below)

---
Strengths:
- The results seem to be promising
- The paper is globally well written.

Weaknesses:
- Contributions: the method and the algorithm do not seem very clear because there are large discrepancies between the few theoretical results and the actual algorithm (see questions and limitations). The claim in the abstract "We provide theoretical analysis on the proposed algorithm." does not seem very accurate.
- Reproducibility: Concerning the code, is it proprietary or do you want to publish it? It seems that at least one of these two claims is wrong (1) "Did you include the license to the code and datasets? [No] The code and the data are proprietary." and (2) "Did you include the code, data, and instructions needed to reproduce the main experimental results (either in the supplemental material or as a URL)? [No] We will publish code once paper is accepted."
- Presentation: The approach is only presented starting from page 6 and it is basically presented in two pages. It would be useful to go more directly and more in depth into the actual contributions of the paper.
- (minor) Typos/grammar: "we randomly sample a weather at each episode starts.", "Deep RL algorithms and attracts more and more attraction in the RL community".instead of "expensive" experiments, I suppose you meant "extensive"? (even though both might be true).

---

> ### Author Response · Authors · 2022-08-09
> **Reply to Reviewer nvyx (1/2)**
>
> Thanks for your constructive comments!
>
> We would like to clarify the contributions of this paper: 1) We firstly discover and analyze the approximation issues of existing behavioral metric-based representation learning for RL (in Section 4) and; 2) we propose a new behavioral metric RAP along with an approximation approach that addresses aforementioned approximation issues (in Section 5).
>
> >  Contributions: the method and the algorithm do not seem very clear because there are large discrepancies between the few theoretical results and the actual algorithm (see questions and limitations).
>
> We will answer your questions in corresponding sections.
>
> > The claim in the abstract "We provide theoretical analysis on the proposed algorithm." does not seem very accurate.
>
> Sorry for the confusion. We formally define our RAP distance (in Definition 5.1), then prove its convergence and the existence of fixed-point (Theorem 5.2), and derive its value function difference bound (Theorem 5.3 with proof in Appendix A.3).
>
> > Reproducibility: Concerning the code, is it proprietary or do you want to publish it? It seems that at least one of these two claims is wrong (1) "Did you include the license to the code and datasets? [No] The code and the data are proprietary." and (2) "Did you include the code, data, and instructions needed to reproduce the main experimental results (either in the supplemental material or as a URL)? [No] We will publish code once paper is accepted."
>
> We will publish our code when this paper is accepted. The sentence "[No] The code and the data are proprietary" in line 483 to line 486 are examples that came with Neurips Latex template. They are NOT our answers. Our answers for the checklist start from line 490.
>
> > Presentation: The approach is only presented starting from page 6 and it is basically presented in two pages. It would be useful to go more directly and more in depth into the actual contributions of the paper.
>
> Because we need pages to analyze the approximation issues for the existing behavioral metrics in order to motivate the proposed RAP distance. This part is also the theoretical contribution to this paper. We will revise section 5 more in depth in the final version where there would be one more page.
>
> > (minor) Typos/grammar: "we randomly sample a weather at each episode starts.", "Deep RL algorithms and attracts more and more attraction in the RL community".instead of "expensive" experiments, I suppose you meant "extensive"? (even though both might be true).
>
> Sorry for the typos and grammar issues. We have revised them in the new version.
>
> >  In Equation 9, d is defined as a distance from a space of distance functions.
> >
> >  * Can you confirm that some distance functions d wouldn't make a lot of sense?
> >
> > * Theorem 5.2 and theorem 5.3 uses the l_\infty loss but then in Section 5.2, a different distance function is used. How are the theorems useful for the practical algorithms?
>
> The definition 5.2 might be confused. The function $\mathcal{F}^{\pi}\_G$ should be defined as the RAP distance "**transformation**" function, which transforms a distance function to another distance function. The $L\_{\infty}$ norm in theorem 5.2 (but not in theorem 5.3) measures the distance between distance function $d$ and $d^{\prime}$, i.e., the $L\_{\infty}$ norm is a distance function of distance functions. The $L\_{\infty}$ norm is a mathematical tool to prove the transformation function $\mathcal{F}^{\pi}\_G$ satisfies Banach’s fixed-point theorem so that $\mathcal{F}^{\pi}\_G$ has a unique fixed-point $d^{\pi}\_G$. Similar proofs for Banach’s fixed-point theorem that use $L\_{\infty}$ have existed in MICo[1] distance Value Iteration[2], and Q-learning.[3]
>
> The unique fixed-point $d^{\pi}\_G = \mathcal{F}^{\pi}\_G(d^{\pi}\_G)$ is the behavioral metric we want to approximate, instead of $L\_{\infty}$.
> Banach’s fixed-point theorem indicates that by continuously iterating $\mathcal{F}^{\pi}\_G$ from a random distance function $d$, eventually it will converge to the fixed-point $d^{\pi}\_G$.
> In section 5.2, we try to approximate $d^{\pi}\_G$, by iterating Eq.(9). It is similar to the DQN, as they iteratively approximate the bellman operator and eventually learn the optimal Q-function.

---

> ### Author Response · Authors · 2022-08-09
> **Reply to Reviewer nvyx (2/2)**
>
> > In addition to the lack of connexion between the theorems and the actual algorithm due to the metric d, an important characteristic/potential limitation of the RAP distance is that it depends on the policy. The explicit dependance on the policy is removed in Equation 13, why? How do you train the algorithm in practice given that samples might not be all obtained with one given known policy?
>
> Equation 13 is dependent on the policy. The expectation of loss (13) is taken over the replay buffer. The rewards, actions, and states are sampled from the replay buffer and generated by the policy and MDP. The learning of approximated reward variance is also dependent on the sampled data from replay buffer. We provide a detailed version of the RAP loss in Eq (21) in Appendix F, which adds the dependence on the replay buffer.
>
> We add a detailed explanation in Appendix F to address this question, with additional experiments that verify the RAP distance learned from off-policy replay buffer would not be far away from on-policy. In the additional experiments, we reduce the replay buffer size so that the RL algorithm is more close to on-policy. Then we compare the RAP distances measured in our approximation (replay buffer size of 1M) to smaller replay buffer sizes (500K, 100K, 50K, and 10K). Table 4 ([https://imgur.com/4c1lZqM](https://imgur.com/4c1lZqM)) in Appendix F shows the differences of distances between each two buffer sizes. It indicates that the proposed method is not far away from the models learned with other buffer sizes. The details of the experiments can be found in Appendix F.
>
> ----
> Reference
>
> [1] Castro et al. MICo: Improved representations via sampling-based state similarity for markov decision processes. Neurips 2021
>
> [2] https://sites.ualberta.ca/~szepesva/papers/RLAlgsInMDPs.pdf
>
> [3] http://users.isr.ist.utl.pt/~mtjspaan/readingGroup/ProofQlearning.pdf

---

### Official Review · Reviewer_EWpZ · 2022-07-10

**Rating:** 5
**Confidence:** 4
**Soundness:** 3 good
**Presentation:** 2 fair
**Contribution:** 2 fair

**Summary:**

This paper proposes a representation learning approach based on behavioral metric. The core idea is to project states into an embedding space such that the distance between them w.r.t behavioral similar is preserved. Therefore any distractors or variability in the environments could be potentially soaked due to the flexibility in the loss function. This is contrary to L2 reconstruction based objectives as auxiliary tasks which will learn to capture task irrelevant details in each variable environmental condition.

The core technical idea is to formulate the behavioral metric based approach as a new loss function. This new loss function is added as an auxiliary loss within an actor critic framework. In order to make the loss function tractable, two core assumptions are made: (1) reward expectations in the loss function of the behavioral metric are simplified to absolute errors in the rewards between two sampled states and (2) a new distance metric (RAP) is proposed, which leverages expectations over future states under the policy instead of expensive wasserstein based metrics over the transition model.

The approach is validated on a variety of continuous control tasks. The main results is that the expected returns is higher with this method over baselines.

**Questions:**

- Figure 3 and 4 have mixed results. It might be helpful to run on a more diverse variety of tasks to learn about strengths and weaknesses

- It might be helpful to analyze and show state similarities in a navigation or some 2D projected view of an environment to better understand behaviorally induced state similarities. Currently it is all hidden under a single empirical performance metric and it is very difficult to understand what is happening.

- "we can learn a neural network approximator to estimate it by assuming that the reward rs on state s is Gaussian distributed.". This seems like a major assumption for many environments with discrete reward functions. What type of environments and problems would break under these assumptions?

- "Similarly, in order to estimate the expected next states... we learn a dynamics model Pb taking input of state ... outputs a Gaussian distribution over the next state embedding". Is this assumption necessary? Can a neural network be used to estimate the successor representation trained via TD learning?


**Limitations:**

Discussed above.

**Strengths And Weaknesses:**

Strengths:

This method uses behavioral metrics for representation learning which can be more robust to environmental variations. It is an interesting and well grounded approach to handle all sorts of variabilities, especially visual distractors. Instead of hand crafting augmentations, this approach could in principle learn more robust policies under arbitrary pixel variations without apriori knowing the set of invariances and equivariances in the environment.

So it might be a better approach where practitioners don't have good intuitions about the domain and therefore making it hard to build good data augmentations.

Weakness:
1. "Our method achieves comparable scores with data augmentation method DrQ, and its score is higher than behavioral metric DBC and MICo." -> This empirical finding suggests to me that the authors should try this method on domains where it is very difficult to design data augmentations manually.

2. While the empirical performance is shown to be better on some environments, it is not clear why it works so well in some cases and not in others. There is no qualitative analysis so it is difficult to understand what this method is really learning.

---

> ### Author Response · Authors · 2022-08-09
> **To Reviewer EWpZ (1/2)**
>
> Thanks for your constructive comments!
>
> > "Our method achieves comparable scores with data augmentation method DrQ, and its score is higher than behavioral metric DBC and MICo." -> This empirical finding suggests to me that the authors should try this method on domains where it is very difficult to design data augmentations manually.
>
> We perform experiments on CARLA domain because it is a benchmark that previous method DBC has also shown in their paper. CARLA is also a simulator that provides more "real-world" scenarios to evaluate the generalization of methods. It would be fair to compare methods in a "real" domain.
> In the experiment section, we also show two tasks, Door Opening and Two Arm Peg-In-Hole, in Robosuite domain where DrQ performs much worse than our RAP. The tasks cheetah-run and walker-walk in DMC with natural video background also show that RAP achieves higher scores than DrQ.
>
> > While the empirical performance is shown to be better on some environments, it is not clear why it works so well in some cases and not in others. There is no qualitative analysis so it is difficult to understand what this method is really learning.
>
> In order to analyze our method, we add an ablation study in Appendix D.1 and the results are shown in Figure 8 ([https://imgur.com/OcCnGaZ](https://imgur.com/OcCnGaZ)). We try to remove the reward variance correction and find that it becomes more unstable and converges to lower scores.
> This experiment verifies the correction of reward variance can stabilize the learning process of RL and improve the final performance.
> We also remove the learned dynamics in the ablation study in Appendix D.1. We find that the performance would decrease a lot if using the "true" next states. This experiment shows that by approximating the expectation of dynamics model RAP can stably learn to better performance. Details of ablation study can be found in  Appendix D.1.
>
> > Figure 3 and 4 have mixed results. It might be helpful to run on a more diverse variety of tasks to learn about strengths and weaknesses
>
> We run an additional experiment on a domain of 2D highway driving task in Appendix D.2.
> The 2D highway task contains a straight road with 4 lanes and 50 other vehicles running on.
> the RL agent learns to drive the vehicle as far as possible within 600 time steps.
> The experimental result in Figure 11 ([https://imgur.com/ZEg1hql](https://imgur.com/ZEg1hql)) shows that DBC, DrQ, and the proposed method RAP achieve similar scores. SAC and MICo are slightly worse than the others. This experiment indicates we have consistent results with DMC. The details of this experiment can be found in Appendix D.2.
>
> > It might be helpful to analyze and show state similarities in a navigation or some 2D projected view of an environment to better understand behaviorally induced state similarities. Currently it is all hidden under a single empirical performance metric and it is very difficult to understand what is happening.
>
> In order to further analyze our method, apart from the ablation study, we add visualizations of representation space in Appendix E (Figure 12-14).
> We compare the t-SNE visualizations between our RAP and other behavioral metric-based representation learning methods, i.e., MICo and DBC.
> Our method RAP can map the observations to a representation space that is able to generalize to the task-irrelevant background (in Figure 12 ([https://imgur.com/MVbj8ph](https://imgur.com/MVbj8ph))). Figure 13 ([https://imgur.com/JblokXs](https://imgur.com/JblokXs)) shows that RAP representation space is more related to the value function which means it preserves more information to predict the Q-value. Figure 14 ([https://imgur.com/LYSLx4M](https://imgur.com/LYSLx4M)) shows that RAP can map similar observations even with different backgrounds to close representations.

---

> ### Author Response · Authors · 2022-08-09
> **To Reviewer EWpZ (2/2)**
>
> > "we can learn a neural network approximator to estimate it by assuming that the reward rs on state s is Gaussian distributed.". This seems like a major assumption for many environments with discrete reward functions. What type of environments and problems would break under these assumptions?
>
> We have two reasons to take the Gaussian distribution assumption for reward function:
> 1. In control tasks, the reward function is usually continuous (real number) and dense.
> 2. In control tasks, the policy is usually formulated as Gaussian stochastic policy with continuous action space. (as how SAC does)
>
> If the tasks have discrete rewards or the action space is discrete, the Gaussian distribution assumption for the reward function may not hold.
> However, the Gaussian distribution is not the necessary assumption in our method. The approximation of RAP requires the estimation of **variance** of reward. If the reward function is discrete and the action space is discrete, we can learn a categorical distribution for the reward function, in order to better estimate the variance of reward. In conclusion, we do not have to assume Gaussian distribution for reward in all tasks. Instead, we can model the reward functions based on domain knowledge of the tasks.
>
> > "Similarly, in order to estimate the expected next states... we learn a dynamics model Pb taking input of state ... outputs a Gaussian distribution over the next state embedding". Is this assumption necessary?
>
> Our additional ablation study shown in Figure 8 ([https://imgur.com/OcCnGaZ](https://imgur.com/OcCnGaZ)) in Appendix D.1 shows that learning a Gaussian distribution for dynamics model can improve performance. The distance measured by the expectation/mean of dynamics model can stabilize the learning of the metric. Using sampled next states, e.g. MICo, may suffer from some randomness induced by the sampling of MDP dynamics.
>
> > Can a neural network be used to estimate the successor representation trained via TD learning?
>
> Many existing methods also use neural network to predict representations of next states using TD learning, such as DBC and DeepMDP[1]. Model-based methods also learn a world model which includes dynamics in latent space, such as PlaNet[2] and Dreamer[3].
>
> ------
>
> Reference
>
> [1] Gelada et al. DeepMDP: Learning Continuous Latent Space Models for Representation Learning. ICML 2019.
>
> [2] Hafner et al. Learning Latent Dynamics for Planning from Pixels. ICML 2019.
>
> [3] Hafner et al. Mastering Atari with Discrete World Models. ICLR 2021.

---

### Official Review · Reviewer_TAKM · 2022-07-11

**Rating:** 7
**Confidence:** 3
**Soundness:** 3 good
**Presentation:** 4 excellent
**Contribution:** 3 good

**Summary:**

In this paper, the authors focus on the behavior metric design problem so as to learn an informative and robust representation. To this end, three theoretical issues of the current behavior metrics are analyzed, including loss function mismatch, relaxation of dynamics model divergence, and limitation of the L1/L2 norm. Then a new metric named Reducing Approximation Gap (RAP) distance is proposed, and a practical representation learning algorithm is proposed based on RAP. Empirical results on three benchmarks show the effectiveness of the proposed method.

**Questions:**

1. What is the tightness of RAP bound in Eq. 10 compared with $\pi$-bisimulation metric [1] and MICo [2]? Can RAP bring a tighter bound?
2. Why the performance curves of baselines like DrQ in Figure 3 are lower than their original papers'? For example, DrQ can get 800 score in the original DrQ paper (Figure 3).
3. More visualization of the latent space property of RAP over $\pi$-bisimulation and MICo is preferred.

[1] Castro P S. Scalable methods for computing state similarity in deterministic markov decision processes[C]//Proceedings of the AAAI Conference on Artificial Intelligence. 2020, 34(06): 10069-10076.

[2] Castro P S, Kastner T, Panangaden P, et al. MICo: Improved representations via sampling-based state similarity for Markov decision processes[J]. Advances in Neural Information Processing Systems, 2021, 34: 30113-30126.

**Limitations:**

No.

**Strengths And Weaknesses:**

**originality**

The main contributions of this paper are clear. The authors carefully analyze the theoretical issue of current behavior metrics. Motivated by the above issues, a novel behavior metric for reducing the approximation gap from the perspective of the dynamics model. A thorough theoretical proof is given for the value function difference bound.

**quality**

The empirical results on three different domains, i.e. DM Control, Robot control, and autonomous driving are given. The performance curves validate the effectiveness of the proposed method. But For some environments, e.g. cheetah_run in DMC, the baseline curves are not consistent with the original paper.

**clarity**

The paper is well-written and easy to follow. Although this is a theoretical paper, the authors give a very clear background and the main problems of the current behavior metrics. These details help to understand what is the main contribution of the proposed RAP and why it gives a better theoretical property. But for the experimental part, more visualization of the learned latent space property will help to understand what is the difference between RAP over $\pi$-bisimulation metric and MICo.

**significance**

The representation learning plays an essential way in reinforcement learning (RL) for achieving generalizable and robust capacity. The behavior metrics-based research has drawn large attention in the RL community. The proposed method not only gives the theoretical shortcomings of the previous metrics but also provides a new metric with the better theoretical property.

---

> ### Author Response · Authors · 2022-08-09
> **To Reviewer TAKM**
>
> Thanks for your comments and summarization.
>
> > 1.What is the tightness of RAP bound in Eq. 10 compared with $\pi$-bisimulation metric [1] and MICo [2]? Can RAP bring a tighter bound?
>
> Unfortunately, we are not able to compare the tightness of the bounds between RAP, $\pi$-bisimulation metric and MICo. We believe they have similar tightness. This paper provides the bound for the approximated distance which is looser than the bound of the original distance. After that, we develop an approximation algorithm in section 5.2 to make the approximated distance closer to the original distance and to have a tighter bound.
>
> > 2. Why the performance curves of baselines like DrQ in Figure 3 are lower than their original papers'? For example, DrQ can get 800 score in the original DrQ paper (Figure 3).
>
> All DrQ results in the experiment section are using the codes which are downloaded from the GitHub repository that DrQ published. We recognize that DrQ paper has different results on cheetah_run. Figure 2 in DrQ paper cheetah_run only gets around 600 scores at 1M steps, which is similar to our results in Figure 3. Even though DrQ can get 800 scores in cheetah run with original background, our method is still comparable. Besides, our method still outperforms DrQ in natural video background setting.
>
> > 3.More visualization of the latent space property of RAP over $\pi$-bisimulation and MICo is preferred.
>
> Thanks for your suggestion. We add a visualization of latent space in Appendix E (Figure 12-14) compared with MICo and DBC. Our method RAP can map the observations to a representation space that is able to generalize to the task-irrelevant background (in Figure 12 ([https://imgur.com/MVbj8ph](https://imgur.com/MVbj8ph))). Figure 13 ([https://imgur.com/JblokXs](https://imgur.com/JblokXs)) shows that RAP representation space is more related to the value function which means it preserves more information to predict the Q-value. Figure 14 ([https://imgur.com/LYSLx4M](https://imgur.com/LYSLx4M)) shows that RAP can map similar observations even with different backgrounds to close representations.

---

### Official Review · Reviewer_BWMd · 2022-07-11

**Rating:** 7
**Confidence:** 4
**Soundness:** 4 excellent
**Presentation:** 3 good
**Contribution:** 3 good

**Summary:**

The paper proposes a novel state distance metric to regularize the latent space of model-free reinforcement learning agents. The proposed RAP metric accounts for the variance of the rewards received in a pair of states under a stochastic policy. This enables the metric to be an upper bound on the difference between the values of a pair of states. The metric is added as a latent space regularizer to the soft actor-critic and evaluated in multiple high-frequency continuous-control environments.

**Questions:**

1. Is it straightforward to implement the RAP metric compared to MICo? Could you include a code snippet to compare the implementation of the two metrics?
2. I would like to see an ablation where the correction for the variance of the rewards in states s_i and s_j is removed. Would the ablation be similar / identical to MICo?

**Limitations:**

Limitations were not addressed. I would like to see a discussion of using the RAP metric with agents other than the soft actor-critic.

**Strengths And Weaknesses:**

Strengths:
* The paper contains a rigorous analysis of previously proposed behavioral metrics to motivate the proposed RAP metric.
* Excellent evaluation in multiple continuous control environments. Strong results in favor of the proposed metric in all domains.

Weaknesses:
* The derivation and interpretation of some equations could be made clearer.
* No analysis of the learned models. E.g. a visualization of the learned latent space should be included if the main contribution of this paper is a regularizer on the latent space.

Comments:

– It is not completely clear how we went from equation 12 to 13. Equation 13 should be described in more than one sentence.

– I believe the term \hat{\sigma(s_i)} is dependent on the policy \pi. Equation 13 might be misleading if the policy is not included in the notation.

– Equations 11 and 12 should be cleaned up. Using colors or other annotations to match the text and the terms in the equation would be helpful.

– Line 193: bad spacing.

---

> ### Author Response · Authors · 2022-08-09
> **Reply to Reviewer BWMd (1/2)**
>
> Thanks for your insightful comments!
>
> > The derivation and interpretation of some equations could be made clearer.
>
> Thanks for pointing out this issue. We will revise the paper carefully and improve the derivation and interpretation.
>
> > No analysis of the learned models. E.g. a visualization of the learned latent space should be included if the main contribution of this paper is a regularizer on the latent space.
>
> We add a visualization of latent space in Appendix E (Figure 12-14). Our method RAP can map the observations to a representation space that is able to generalize to the task-irrelevant background (in Figure 12 ([https://imgur.com/MVbj8ph](https://imgur.com/MVbj8ph))). Figure 13([https://imgur.com/JblokXs](https://imgur.com/JblokXs)) shows that RAP representation space is more related to the value function which means it preserves more information to predict the Q-value. Figure 14([https://imgur.com/LYSLx4M](https://imgur.com/LYSLx4M)) shows that RAP can map similar observations even with different backgrounds to close representations.
>
> > It is not completely clear how we went from equation 12 to 13. Equation 13 should be described in more than one sentence.
>
> Due to the page limits, we would like to explain Eq.(13) here for you. A simple way to learn Eq.12 is to minimize the error between the left side and the right side. However, the expectation under the square root will introduce new bias in learning. We move the dynamics difference term to the left,
>
> $$
>  d^{\pi}\_G(\mathbf{s}\_i, \mathbf{s}\_j) - \gamma \mathbb{E}\_{\substack{\mathbf{a}\_i \sim \pi, \mathbf{a}\_j \sim \pi}}  d^{\pi}\_G( \mathbb{E}\_{{s}^{\prime}\_i \sim P^{\mathbf{a}\_i}\_{\mathbf{s}\_i}}[{s}^{\prime}\_i],  \mathbb{E}\_{{s}^{\prime}\_j \sim P^{\mathbf{a}\_j}\_{\mathbf{s}\_j}}[{s}^{\prime}\_j])
> =  \sqrt{ \mathbb{E}\_{\substack{\mathbf{a}\_i \sim \pi , \mathbf{a}\_j \sim \pi}} \left[ \left|r^{\mathbf{a}\_i}\_{\mathbf{s}\_i} - r^{\mathbf{a}\_j}\_{\mathbf{s}\_j} \right|^2 \right] - Var[r\_{\mathbf{s}\_i}] - Var [ r\_{\mathbf{s}\_j}] } .
> $$
>
> Then we take square on both sides and get
>
> $$
> (d^{\pi}\_G(\mathbf{s}\_i, \mathbf{s}\_j) - \gamma \mathbb{E}\_{\substack{\mathbf{a}\_i \sim \pi, \mathbf{a}\_j \sim \pi}}  d^{\pi}\_G( \mathbb{E}\_{{s}^{\prime}\_i \sim P^{\mathbf{a}\_i}\_{\mathbf{s}\_i}}[{s}^{\prime}\_i],  \mathbb{E}\_{{s}^{\prime}\_j \sim P^{\mathbf{a}\_j}\_{\mathbf{s}\_j}}[{s}^{\prime}\_j]) )^2
> =   \mathbb{E}\_{\substack{\mathbf{a}\_i \sim \pi , \mathbf{a}\_j \sim \pi}} \left[ \left|r^{\mathbf{a}\_i}\_{\mathbf{s}\_i} - r^{\mathbf{a}\_j}\_{\mathbf{s}\_j} \right|^2 \right] - Var[r\_{\mathbf{s}\_i}] - Var [ r\_{\mathbf{s}\_j}] .
> $$
>
> Eq.(13) replaces the distance $d$ to the deep network approximated distance  $\hat{d}$ and minimizes the error between the left side and the right side.
>
> > I believe the term \hat{\sigma(s_i)} is dependent on the policy \pi. Equation 13 might be misleading if the policy is not included in the notation.
>
> $\hat{\sigma}(s)$ is a neural network function approximating standard deviation $\sqrt{Var[r_{\mathbf{s}}]}$. The learning of $\hat{\sigma}(s)$ is shown in Appendix B.1.2. It is learned from the data, states and rewards, sampled from replay buffer. Since the reward is received after taking an action, the learning of the reward function including $\hat{\sigma}(s)$ is dependent on the policy in replay buffer.
>
> For Eq.(13), the expectation is taking over a batch of transitions sampled from replay buffer, which are the same data for the learning of SAC (as shown in Algorithm 1 in Appendix B.2). Eq.(13) can be considered dependent on the policy. We will revise Eq.(13) to clarify it.
>
> >  Equations 11 and 12 should be cleaned up. Using colors or other annotations to match the text and the terms in the equation would be helpful.
>
> Thank you very much for the suggestion. We will add more annotations to Eq. (11) and (12). But currently due to the limited pages, we have no space to do so. We promise we will revise section 5.2 carefully to make it more clear to the readers.
>
> >  Line 193: bad spacing.
>
> Thanks for pointing it out. We will revise the spacing.

---

> ### Author Response · Authors · 2022-08-09
> **Reply to Reviewer BWMd (2/2)**
>
> > 1.Is it straightforward to implement the RAP metric compared to MICo? Could you include a code snippet to compare the implementation of the two metrics?
>
> It is possible to implement RAP upon MICo by adding the modules for learning rewards variance and dynamics mean. However, we use different framework to implement our RAP instead of modifying codes from MICo implementation. We will open-source our code once this paper is accepted.
>
> > 2.I would like to see an ablation where the correction for the variance of the rewards in states s_i and s_j is removed. Would the ablation be similar / identical to MICo?
>
> We add an ablation study in Appendix D.1. We try to remove the reward variance correction and find that it becomes more unstable and converges to lower scores. This ablation is not identical to MICo. In MICo, the next states for measuring metrics are sampled from "real" MDP. But in RAP we learn a dynamics model to approximate the dynamics in MDP, and compute the metric only with the expectation (mean) of the learned dynamics. We also remove the learned dynamics in the ablation study in Appendix D.1. We find that the performance would decrease a lot if using the "true" next states. The result of ablation is shown in Figure 8 ([https://imgur.com/OcCnGaZ](https://imgur.com/OcCnGaZ)). Details of ablation study can be found in  Appendix D.1.

---

### Author Response · Authors · 2022-08-02
**We will update soon**

Due to the authors getting covid-19 positive in the past few days, some additional experiments have not been finished. We will update the reviewers soon.

---

### Meta-Review · Area_Chair_jjr5 · 2022-08-28

**Recommendation:** Accept
**Confidence:** Certain

**Metareview:**

Unanimous accept from 4 reviewers

This paper was initially divisive among reviewers (scoring 7744), now 7756 post rebuttal

While the consensus of this representation learning work had initial strengths of clear analysis of previous metrics, clear evaluations, writing, and good experimentation; the prior weaknesses were mainly confusion about derivations and equation interpretations, no analysis of learned model, no visualizations of the latents space, lack of explanation why the method only does well in some environments and not others. These confusions have since been cleared up, in particular the authors have since added additional analysis in the appendix is useful (e.g. Table 4 and Fig 15) to satisfy reviewer's nvyx's concerns over clarifications about equations

**Award:**

No

---

### Decision · Program_Chairs · 2022-09-14

Accept